# Single molecule microscopy reveals key physical features of repair foci in living cells

Judith Miné-Hattab[1]*, Mathias Heltberg[1,2], Marie Villemeur[1], Chloé Guedj[1], Thierry Mora[2], Aleksandra M Walczak[2], Maxime Dahan[3†], Angela Taddei[1,4]*

[1]Institut Curie, PSL University, Sorbonne Université, CNRS, Nuclear Dynamics, Paris, France; [2]Laboratoire de Physique de l'Ecole Normale Supérieure, PSL University, CNRS, Sorbonne Université , Université de Paris, Paris, France; [3]Institut Curie, PSL University, Sorbonne Université, CNRS, Physico Chimie Curie, Paris, France; [4]Cogitamus Laboratory, Paris, France

**Abstract** In response to double strand breaks (DSB), repair proteins accumulate at damaged sites, forming membrane-less sub-compartments or foci. Here we explored the physical nature of these foci, using single molecule microscopy in living cells. Rad52, the functional homolog of BRCA2 in yeast, accumulates at DSB sites and diffuses ~6 times faster within repair foci than the focus itself, exhibiting confined motion. The Rad52 confinement radius coincides with the focus size: foci resulting from 2 DSBs are twice larger in volume that the ones induced by a unique DSB and the Rad52 confinement radius scales accordingly. In contrast, molecules of the single strand binding protein Rfa1 follow anomalous diffusion similar to the focus itself or damaged chromatin. We conclude that while most Rfa1 molecules are bound to the ssDNA, Rad52 molecules are free to explore the entire focus reflecting the existence of a liquid droplet around damaged DNA.

*For correspondence:
judith.Mine@curie.fr (J);
angela.taddei@curie.fr (AT)

†Deceased.

## Introduction

The cell nucleus contains membrane-less sub-compartments inside which specific proteins are more concentrated than elsewhere in the nucleus (*Meldi and Brickner, 2011*; *Miné-Hattab and Taddei, 2019*; *Rippe, 2007*). Such regions of high local protein concentration, called 'foci', are hypothesized to help proteins coordinate and collectively perform their function (*Banani et al., 2017*; *Cabianca et al., 2019*; *Chubb and Bickmore, 2003*; *Hübner and Spector, 2010*; *Miné-Hattab and Taddei, 2019*). The formation of these foci at the right place in the nucleus and within a well-defined time window is essential for the functioning of the cell. Here, we focus on repair sub-compartments formed in response to double strand breaks (DBS) (*Lisby et al., 2004*). DNA repair is an essential process for preserving genome integrity. Among the different kinds of DNA damages, DSBs are the most genotoxic (*Iliakis et al., 2004*). Failure to repair such lesions leads to genomic instability or cell death, and mutations in DNA repair genes lead to diseases such as Werner, Bloom and other cancer predisposition syndromes (*Rassool, 2003*; *Thorslund and West, 2007*). Eukaryotic organisms use mainly two major mechanisms to repair DSBs: non-homologous end-joining (NHEJ) and homologous recombination (HR) (*Wyman and Kanaar, 2006*). HR occurs primarily in S/G2 phase cells and uses an undamaged homologous DNA sequence as a template for copying the missing information (*Lisby and Rothstein, 2015*). The biochemistry and the genetics of DSB repair by HR have been extensively investigated *in vitro* (*Kaniecki et al., 2018*; *Miné et al., 2007*) and *in vivo* (*Lisby and Rothstein, 2009*; *Sánchez et al., 2017*). HR is orchestrated by mega-Dalton multi-protein complexes of 500 to 2000 proteins that colocalize with the DSB (*Lisby et al., 2001*). In living cells, these protein centers can be visualized as fluorescent foci using tagged HR proteins. Among the proteins

occupying these centers are the enzymes of the highly conserved Rad52 epistasis group, including Rad51, Rad52 and Rad54 (*Symington et al., 2014*). When a DSB forms, the 5′ ends of the DNA break are resected by nucleases to yield 3′ single-stranded DNA (ssDNA) tails that are rapidly coated by the RPA complex. The Rad52 protein, the functional analog of human BRCA2 in yeast, then stimulates the removal of RPA and recruits the recombinase Rad51 to the ssDNA tail on which it polymerizes (*Symington et al., 2014*). The Rad51-ssDNA complex, called a nucleo-filament, has the capacity to search and identify a region of homology and to promote strand invasion of the homologous duplex DNA (*Bordelet and Dubrana, 2019*). Once homology is found, the invading strand primes DNA synthesis of the homologous template, ultimately restoring genetic information disrupted at the DSB. How repair foci and more generally membrane-less sub-compartments are formed, maintained and disassembled at time scales relevant for their biological function remains unknown. Several models are intensively debated in the literature to understand the nature of membrane-less sub-compartments (*Erdel and Rippe, 2018*; *Hyman et al., 2014*; *McSwiggen et al., 2019a*; *Miné-Hattab and Taddei, 2019*; *Strom et al., 2017*). In particular, recent works proposed that repair proteins exhibit some properties of liquid-like droplets such as Rad52 in *Saccharomyces cerevisiae* yeast (*Oshidari et al., 2020*) and 53BP1 in human cells (*Altmeyer et al., 2015*; *Kilic et al., 2019*; *Pessina et al., 2019*).

Here, we present new insights into the dynamics and the nature of repair foci using Single Particle Tracking (SPT) and Photo Activable Localization Microscopy (PALM) in *Saccharomyces cerevisiae* cells. Watching how proteins move and interact within a living cell is crucial for better understanding their biological mechanisms. SPT is a powerful technique that makes these observations possible by taking 'live' recordings of individual molecules in a cell at high temporal and spatial resolution (50 Hz, 30 nm) (*Dolgin, 2019*; *Manley et al., 2008*; *Oswald et al., 2014*). Based on the way individual molecules move *in vivo*, SPT allows for (i) sorting proteins into subpopulations characterized by their apparent diffusion coefficients, (ii) quantifying their motion, (iii) estimating residence times in specific regions of the nucleus, (iv) and testing the existence of a potential attracting or repelling molecules within distances smaller than the diffraction limit. To complement this approach, PALM allows for measuring the position of molecules at 30 nm resolution to establish density maps of the molecules of interest.

Using SPT and PALM in the presence or absence of DSB, we have assessed the dynamics, for the first time at the single molecule level, of 2 repair proteins: Rad52 and the ssDNA-binding protein Rfa1, a subunit of the RPA complex. We find that inside repair foci, Rad52 molecules are surprisingly mobile: they diffuse an order of magnitude faster than Rfa1, the whole focus itself, and damaged DNA, indicating that most of the Rad52 molecules are not bound to damaged DNA. Instead, Rad52 explores the volume of the focus through confined motion and exhibits a sharp change of diffusion coefficient when entering or escaping foci. In response to multiple DSBs, Rad52 form larger foci than the ones formed upon a single DSB. When foci size is larger, the confinement radius of individual Rad52 inside the focus scales with focus size. Our statistical analysis of single molecule trajectories reveals the existence of an attractive potential maintaining Rad52 molecules within a focus. Altogether, our results using single molecule microscopy indicate that while Rfa1 diffuses similarly to damaged DNA, Rad52 motion exhibits physical properties consistent with diffusion within a Liquid-Liquid Phase Separated droplet.

## Results

### Rad52 exhibits two diffusive behaviors in the absence of DSB

We first investigated the mobility of individual Rad52 molecules *in vivo* in the absence of DNA damage by SPT. To image Rad52 without altering its endogenous expression level, we generated haploid cells expressing the endogenous *Rad52* fused to Halo (*Figure 1A*, *Figure 1—figure supplement 1* and Materials and methods). Prior to visualization on a PALM microscope (see Materials and methods), exponentially growing cells were incubated with fluorescent and fluorogenic JF646, a dye emitting light only once bound to Halo (*Grimm et al., 2015*). We used a low concentration of JF646 allowing for the observation of individual molecules (*Ranjan et al., 2020*; *Figure 1—figure supplement 2*). Rad52-Halo bound to JF646 (Rad52-Halo/JF646) were visualized at 20 ms time intervals (50 Hz) in 2-dimensions during 1000 frames until no signal was visible. A typical

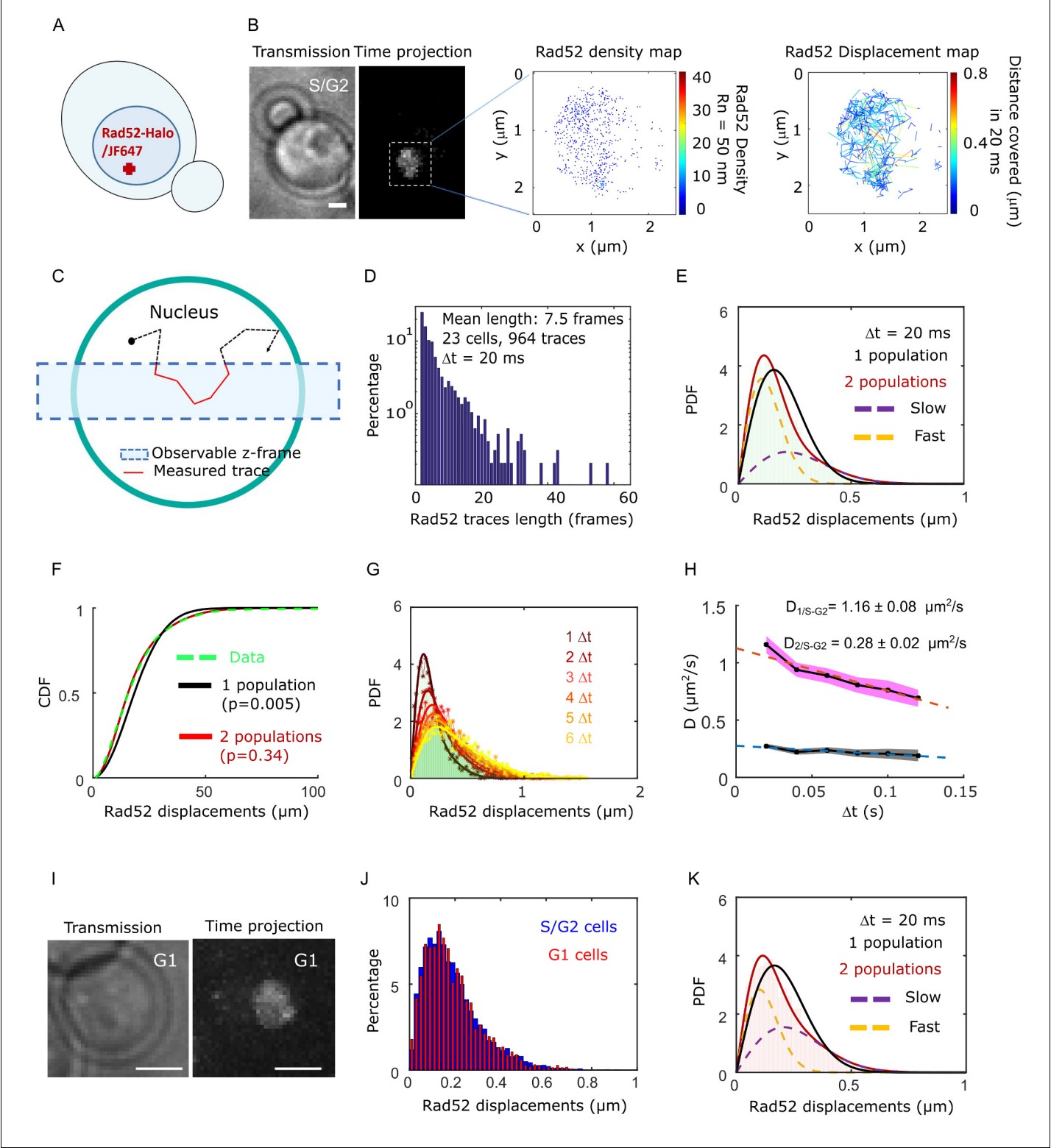

**Figure 1.** Rad52 diffusion observed at the single molecule level, in the absence of DNA damage. (**A**) Design of the experiment. Haploid yeast strain harboring Rad52 endogenously fused to Halo and incubated with fluorogenic JF646 dyes are visualized by Single Particle Tracking (SPT). Individual Rad52-Halo/JF646 are tracked at 20 ms time intervals (50 Hz), movies are 1000 frames long. (**B**) Typical S/G2 haploid cells harboring Rad52-Halo coupled with JF646 (Rad52-Halo/JF646). From left to right: transmission image; time-projection of a typical SPT acquisition (see Materials and methods); Rad52 detections: each spot represents a single detection of Rad52-Halo/JF646, the color map indicates the number of Rad52 neighbors inside a 50

*Figure 1 continued on next page*

*Figure 1 continued*

nm radius disk; Rad52 traces: each line represents the trajectory of a detection, the color map indicates the distance covered in 20 ms. The bar scale represents 1 μm. This typical nucleus exhibits 682 detections and 129 traces. (**C**) Schematic of SPT experiment: individual molecules are tracked in 2-dimensions. The blue rectangle represents the observable z-section. (**D**) Distribution of tracks length of Rad52-Halo/JF646 in the absence of DNA damage. The histogram combines 23 S/G2 phase cells representing 964 traces (mean length of 7.5 fames), and 6252 displacements of 20 ms. S/G2 cells were selected based on the presence and the size of a bud on the transmission image. (**E**) Probability Density Function (PDF) of Rad52-Halo/JF646 molecules of haploid S/G2-phase cells in the absence of DNA damage. The time interval is 20 ms. Red: 2-population fit; dashed purple: slow population; dashed orange: fast population; Black: 1-population fit. (**F**) Cumulative Density Function (CDF) of Rad52-Halo/JF646 molecules in haploid S/G2-phase cells in the absence of DNA damage. Dashed green line: data; Red: 2-population fit; Black: 1-population fit. The p-values are indicated in parenthesis (see Materials and methods). (**G**) Probability Density Function (PDF) of Rad52-Halo/JF646 molecules in haploid S/G2-phase cells in the absence of DNA damage, for time-points spaced by 20, 40, 60, 80, 100 and 120 ms. The lines show the 2-population fit performed on each PDF. (**H**) Diffusion coefficient D (μm²/s) obtained from the fits shown in **G** for each Δt. The fits (dotted lines) represent the expected fraction calculated from simulations (see Materials and methods). (**I**) Typical cell harboring Rad52-Halo/JF646. Left: transmission image; right: time-projection of a typical SPT acquisition (1000 frames). (**J**) Displacement histogram of S/G2 cells (blue) *versus* G1 cells (red). (**K**) Probability Density Function (PDF) of Rad52-Halo/JF646 molecules in haploid G1 cells in the absence of DNA damage. Black: 1-population fit; Red: 2-population fit; dashed purple: slow molecule; dashed orange: fast molecules.

The online version of this article includes the following figure supplement(s) for figure 1:

**Figure supplement 1.** Dilution assay of strains harboring Rad52-mMaple or Rad52-Halo on MMS plates.
**Figure supplement 2.** Visualization of Rad52 at the single molecule level.
**Figure supplement 3.** Characterization of JF646 photo-bleaching in budding yeast.
**Figure supplement 4.** In the absence of DSB, Rad52 explores the whole nucleus.
**Figure supplement 5.** Displacement of free Halo-NLS.

individual cell is shown in *Figure 1B*. After detection and tracking (see Materials and methods), we calculated density and displacement maps of Rad52 molecules (*Figure 1B*). Since Rad52 tracking is performed in 2-dimensions, molecules are observable as long as they stay within the focal plan representing a z-section of about 400 nm (*Hansen et al., 2018*; *Figure 1C*). After examining 23 cells in S/G2 phase of the cell cycle as assessed by the presence and the size of a bud on the transmission image, we obtained an exponentially decreasing distribution of trace lengths with an average trajectory length of 7.5 frames, the longest trace reaching 65 frames (1.3 s) (*Figure 1D*). Since the half-life time of JF646 is 2.1 s (*Figure 1—figure supplement 3*), the short trace lengths observed here are due to molecules moving out of the observable z-section and not the photo bleaching of the JF646 dyes.

To estimate the apparent diffusion coefficient of Rad52 in the absence of DSB, we calculated their displacement histogram (*Hansen et al., 2017*; *Klein et al., 2019*; *Stracy and Kapanidis, 2017*). This method allows us to test whether Rad52 molecules exhibit a single diffusive regime across the nucleus or if they exist in several subpopulations characterized by distinct diffusion coefficients (see Materials and methods). By fitting the displacement histogram with 1- and 2-population fits, we observed that Rad52 exhibits two distinct diffusive behaviors (*Figure 1E–F*, p=0.0001 and p=0.67, two-sided Kolmogorov-Smirnoff (KS) test for the 1- and 2-population fits respectively, see Materials and methods). We obtained $D_{1/S-G2} = 1.16 \pm 0.08$ μm²/s, $D_{2/S-G2} = 0.28 \pm 0.02$ μm²/s as the best-fit diffusion coefficients (see Materials and methods), with the fraction of slow molecules being $0.62 \pm 0.04$. To accurately estimate the apparent diffusion coefficient of Rad52, we then calculated the displacement histograms for several frame rates (20, 40, 60, 80 and 100 ms time intervals) and fitted these experimental histograms with a 2-population fit (*Figure 1G*). When mobility is observed at larger time intervals, mean square displacement per unit time decreases, which stems from the confinement of molecules within the nuclear space (we checked that confined diffusion explained the trend better than anomalous diffusion, see *Figure 1—figure supplement 4*). Fitting these points with a straight line, we extracted the diffusion coefficients to be $D_{1/S-G2} = 1.16 \pm 0.06$ μm²/s, $D_{2/S-G2} = 0.29 \pm 0.02$ μm²/s, which agree with the measurements obtained from the displacement histogram with Δt = 20 ms (*Figure 1G–H*).

To investigate the origin of the two Rad52 populations observed in the nucleus in the absence of DBS, we first asked whether the existence of a slower population of Rad52 is due to its transient association with RPA present at replication forks during S phase. To test this hypothesis, we measured Rad52 mobility in G1 cells. We obtained similar displacement histograms and diffusion coefficients in G1 as in S/G2 cells (*Figure 1I, J and K*, $D_{1,G1} = 1.08 \pm 0.07$ μm²/s, $D_{2,G1} = 0.27 \pm 0.03$ μm²/

s) with similar proportions. Thus, Rad52 diffusion does not change significantly during the cell cycle and the two observed Rad52 populations cannot be simply explained by free *versus* Rad52 molecules transiently associated with the replication fork. We then examined if the two observed populations could correspond to the monomeric and multimeric forms of Rad52 observed *in vitro* (*Saotome et al., 2018*; *Shinohara et al., 1998*). We calculated the predicted diffusion coefficient of Rad52 monomers and multimers using the Stokes-Einstein equation (see Materials and methods). For that, we first estimated the dynamic viscosity inside yeast nuclei by measuring the mobility of free NLS-Halo/JF646 (*Figure 1—figure supplement 5*). We found a diffusion coefficient of $D_{freeHalo-NLS}$ = 1.90 ± 0.06 μm$^2$/s, consistent with previous studies by FRAP (*Larson et al., 2011*). From this measurement, we extracted a dynamic viscosity in the yeast nucleus of 0.122 ± 0.004 Pa.s. Using this measured viscosity, the predicted diffusion coefficients of Rad52 monomers and multimers are $D_{monomer}$ = 1.20 ± 0.04 μm$^2$/s and $D_{multimer}$ = 0.28 ± 0.01 μm$^2$/s (see Materials and methods), consistent with the two populations $D_1$ = 1.16 ± 0.08 and $D_2$ = 0.28 ± 0.02 obtained experimentally. Thus, our measurements of Rad52 diffusion by SPT are consistent with the existence of a mixed population of Rad52 inside nuclei,~2/3 being multimers.

## Inside foci, Rad52 exhibits a slower diffusion coefficient and a confined motion

We next investigated how Rad52 diffusion is affected in response to DSB. A single DSB was induced at an I-*Sce*I target site inserted at the *LYS2* locus in a strain harboring Rad52-Halo (*Figure 2A* and Materials and methods). Of note, in this setting one can follow only the first step of HR, as no donor sequence is available in the genome. Cells were observed 2 hr after inducing the expression of the endonuclease I-*Sce*I driven by the inducible *GAL* promoter. In these conditions, the I-*Sce*I site is cleaved in 95% of the cells (*Batté et al., 2017*) leading to the formation of Rad52 foci (*Miné-Hattab and Rothstein, 2012*) but D-loop extension is not started yet (*Barzel and Kupiec, 2008*; *Piazza et al., 2019*). During the second hour of DSB induction, fluorogenic JF646 were added similarly to the previous experiment without DSB and cells were imaged with the same illumination conditions as in the absence of DSB. Inside foci, we observe at most 1 detection of Rad52-Halo/JF646 per frame during the movie, a necessary condition to avoid mislinking when tracking single molecules (*Figure 1—figure supplement 2*). Unlike the experiment in the absence of DSB, the Rad52 density map exhibits a strong accumulation of molecules corresponding to the repair focus in G2/S cells. S/G2 cells harboring a Rad52 focus were selected for further analysis (*Figure 2B*). As shown by the displacement map (*Figure 2B*, right panel), molecules localized in Rad52 foci cover shorter distances in a 20 ms step; since the tracking is performed in two dimensions, we can observe molecule for a longer time before they move out of the focal plane. Thus, following DSB, the histogram of trace lengths presents a longer tail than before damage (*Figure 2C* and *Figure 2—figure supplement 1*).

To estimate the apparent diffusion coefficient of Rad52 in the presence of a single DSB, we tested a 1-, 2- and 3-population fit of the Rad52 displacement histograms (*Figure 2D*). For the three populations, we obtained the best fitted values of $D_1$ = 1.17 ± 0.06, $D_2$ = 0.24 ± 0.04 and finally $D_3$ = 0.054 ± 0.007 μm$^2$/s (*Figure 2E*, p=0.0001, p=0.27 and p=0.67, two-sided Kolmogorov-Smirnoff (KS) test for the 1-, 2- and 3-population fits respectively). Only the 3-population fit is not rejected. In addition, it allows for recovering the same two populations previously observed in the absence of DSB, but supplemented by a third slow population (with diffusivity $D_3$). The value of $D_3$ obtained in this way is overestimated because of detection noise (for diffusion coefficients $D < 0.1$ μm$^2$/s, we use mean square displacement analysis allowing us to subtract the noise, see Methods and below). A quick inspection of the spatial distribution of step sizes in the nuclear space indicates that the slowest population is found in the zone where Rad52 molecules are dense, that is within the repair focus (*Figure 2B*, displacement map). Using a density threshold to separate the trajectories located within the focus from the rest of the nucleus (*Figure 2F*, left), we then calculated the displacement histograms of the corresponding trajectories (*Figure 2F*, right). We confirm that Rad52 molecules outside foci exhibit similar displacements as in the absence of damage, whereas Rad52 molecules inside the focus are less mobile (*Figure 2F*, right panel). We also confirmed that the diffusion of Rad52 molecules inside foci is well described by a 1-population fit (see Methods and *Figure 2—figure supplement 2*). To accurately estimate the diffusion coefficient of Rad52 inside the focus accounting for detection noise, and also to measure the nature of Rad52 motion, we used mean

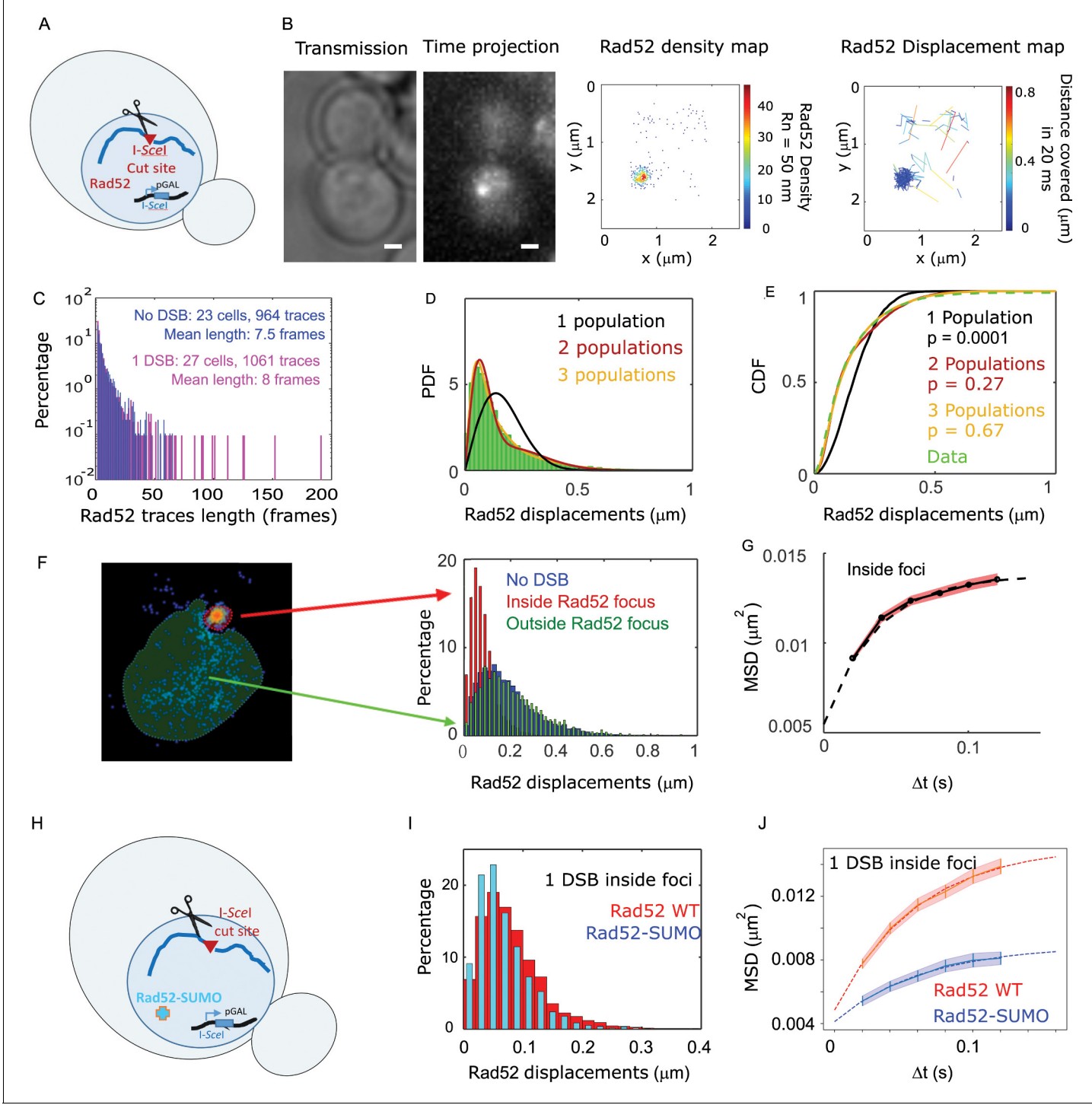

**Figure 2.** Rad52 diffusion at the single molecule level in the presence of a single I-*Sce*I induced DSB. (**A**) Design of the experiment. We induced a single DSB in haploid cells harboring Rad52 endogenously fused to Halo as well as an I-*Sce*I cut site inducible under galactose promoter. The DSB is induced for 2 hours and fluorgenic JF646 dyes are added to the medium during the last hour of incubation prior visualization by SPT.for 2 hr and fluorogenic JF646 dyes are added to the medium during the last hour of incubation prior visualization by SPT. Individual Rad52-Halo/JF646 are tracked at 20 ms time intervals (50 Hz), during 1000 frames. Only cells harboring a Rad52 focus are analyzed. (**B**) Typical S/G2-phase haploid cells harboring Rad52-Halo/JF646. From left to right: transmission image; time-projection of a typical SPT acquisition; Rad52 detections: each spot represents a single detection of Rad52-Halo/JF646, the color map indicates the number of Rad52 neighbors inside a 50 nm radius disk; Rad52 traces: each line represents the trajectory of a detection, the color map indicates the distance in μm covered in 20 ms. The bar scale represents 1 μm. This particular nucleus exhibits 682 detections and 129 traces. (**C**) Distribution of tracks length of Rad52-Halo/JF646 in the presence and the absence of a single DSB. The

*Figure 2 continued on next page*

*Figure 2 continued*

histogram combines 27 S/G2 phase cells, all of them harboring a Rad52 focus, representing 1061 traces (mean length of 8 fames), and 8495 displacements of 20 ms time-intervals. (D) Probability Density Function (PDF) of Rad52-Halo/JF646 molecules in haploid S/G2-phase cells in the presence of a single DSB. The time interval is 20 ms. Green: Rad52 data (27 cells, 1061 trajectories); Black: 1-population fit; Red: 2-population fit; Yellow: 3-population fit. (E) Cumulative Density Function (CDF) of Rad52-Halo/JF646 molecules in haploid S/G2-phase cells in the presence of single DSB. Dashed green line: data; Black: 1-population fit; Red: 2-population fit; Yellow: 3-poulations fit. The p-values are indicated in parenthesis (see Materials and methods). (F) Left: density map of a typical nucleus following the induction of a single DSB. The nucleus is divided in two zones based on a density threshold: the Rad52 focus (highlighted in red) and the rest of the nucleus (highlighted in blue). Right: displacements histograms of trajectories contained inside foci (red) *versus* outside foci (green). If a trajectory crosses the focus boundary, it is cut into two parts: the part inside and the part outside of the focus. The blue histogram represents the displacement of Rad52 molecules in the absence of DSB (shown in *Figure 1E*). (G) Mean Square Displacement (MSD) curve of Rad52/Halo/JF646 molecules inside foci. The dotted line shows a fit of the MSD with a confined model (see Materials and methods). (H) Effect of SUMOylation: we induced a single DSB in haploid cells expressing Rad52 endogenously fused to SUMO and Halo, as well as an I-*Sce*I cut site at *LYS2* locus inducible under galactose promoter. The DSB is induced for 2 hr and fluorogenic JF646 dyes are added to the medium during the last hour of incubation prior to visualization by SPT. Individual Rad52-Halo/JF646 are tracked at 20 ms time intervals (50 Hz), during 1000 frames. (I) Displacement histogram of traces inside foci of Rad52 wild type cells *versus* cells expressing Rad52-SUMO in S/G2 phase cells. A single I-*Sce*I DSB is induced for 2 hr, and only cells harboring a Rad52 focus are analyzed. Red: Rad52 wild type; Blue: Rad52-SUMO. (J) MSD curves of individual Rad52 molecules inside foci (red, same data as *Figure 3E*) *versus* individual Rad52-SUMO molecules inside foci (pink). MSD curves are fitted with a confined model (see Materials and methods).

The online version of this article includes the following figure supplement(s) for figure 2:

**Figure supplement 1.** Cumulative distribution of Rad52-Halo/JF646 traces length in the nucleus.
**Figure supplement 2.** Criteria to determine traces inside repair foci.
**Figure supplement 3.** PDF of Rad52-SUMO after 2 hr of DSB induction Rad52-SUMO is tracked at 20 ms time intervals in cells following 2 hr of DSB induction.
**Figure supplement 4.** Intensity of Rad52 *versus* Rad52-Sumo foci To compare the intensity of Rad52 and SUMOylated Rad52 foci, we observed cells expressing Rad52-Halo of Rad52-SUMO-Halo using wide field microscopy.
**Figure supplement 5.** Mobility of individual Rad52 molecules in haploid *versus* diploid yeast To investigate Rad52 mobility in a situation where the DSB can be repaired, we compared the mobility of individual Rad52 molecules in haploid *versus* diploid cells.

---

squared displacement (MSD) analysis (see Methods). The MSD quantifies the amount of space a molecule explores, and its shape versus time reveals the nature of its movement. This analysis revealed that Rad52 molecules inside foci exhibit a confined motion with a confinement radius $R_c = 124 \pm 3$ nm and a diffusion coefficient $D_{Rad52,inside} = 0.032 \pm 0.006$ $\mu m^2$/s (see Methods and *Figure 2G*).

## Constitutive Rad52 SUMOylation regulates its dynamics within foci

The post-translational modification by the small ubiquitin-like modifier (SUMO) protein has been proposed as a key factor regulating the composition and formation of membrane-less compartments (*Banani et al., 2017*). Furthermore, several studies suggest that this modification plays an important role in mitotic and meiotic recombination (*Altmannova et al., 2010*; *Esta et al., 2013*; *Torres-Rosell et al., 2007*). We thus tested the observables accessible by SPT in a strain expressing Rad52 fused to SUMO. Despite being mildly sensitive to MMS and γ-Rays (*Esta et al., 2013*), Rad52 SUMOylation has several effects *in vitro* and *in vivo*. In particular, SUMOylated Rad52 exhibits lower affinity towards ssDNA and dsDNA, has reduced single-strand annealing activity (*Altmannova et al., 2010*), and partially destabilizes Rad51 filaments compensating the effect of Srs2 deletion (*Esta et al., 2013*). Using a strain expressing Rad52 fused to SUMO and Halo, as well as a single I-*Sce*I cut-site at *LYS2* (*Figure 2H*), we measured the mobility of Rad52-SUMO after 2 hr of DSB induction in cells harboring a focus. We found that Rad52 mobility is significantly slower in the SUMO version of Rad52 than in the wild type (p=1, 2-sided KS test, *Figure 2I*, *Figure 2—figure supplement 3*). As reported in *Silva et al., 2016*, *in vivo*, foci intensity remains similar in SUMOylated Rad52 (*Figure 2—figure supplement 3*). Inside foci, SUMOylated Rad52 also exhibits confined diffusion although with slower diffusion coefficient ($D_{Rad52-SUMO, inside} = 0.009 \pm 0.002$ $\mu m^2$/s, *Figure 2J* and *Supplementary file 1*) and a smaller confinement radius than wild type Rad52 ($Rc_{Rad52-SUMO} = 83 \pm 12$ nm *versus* $Rc_{Rad52} = 124 \pm 3$ nm). Overall, SUMOylated foci are smaller, denser than in wild type cells, and molecules inside foci are less mobile.

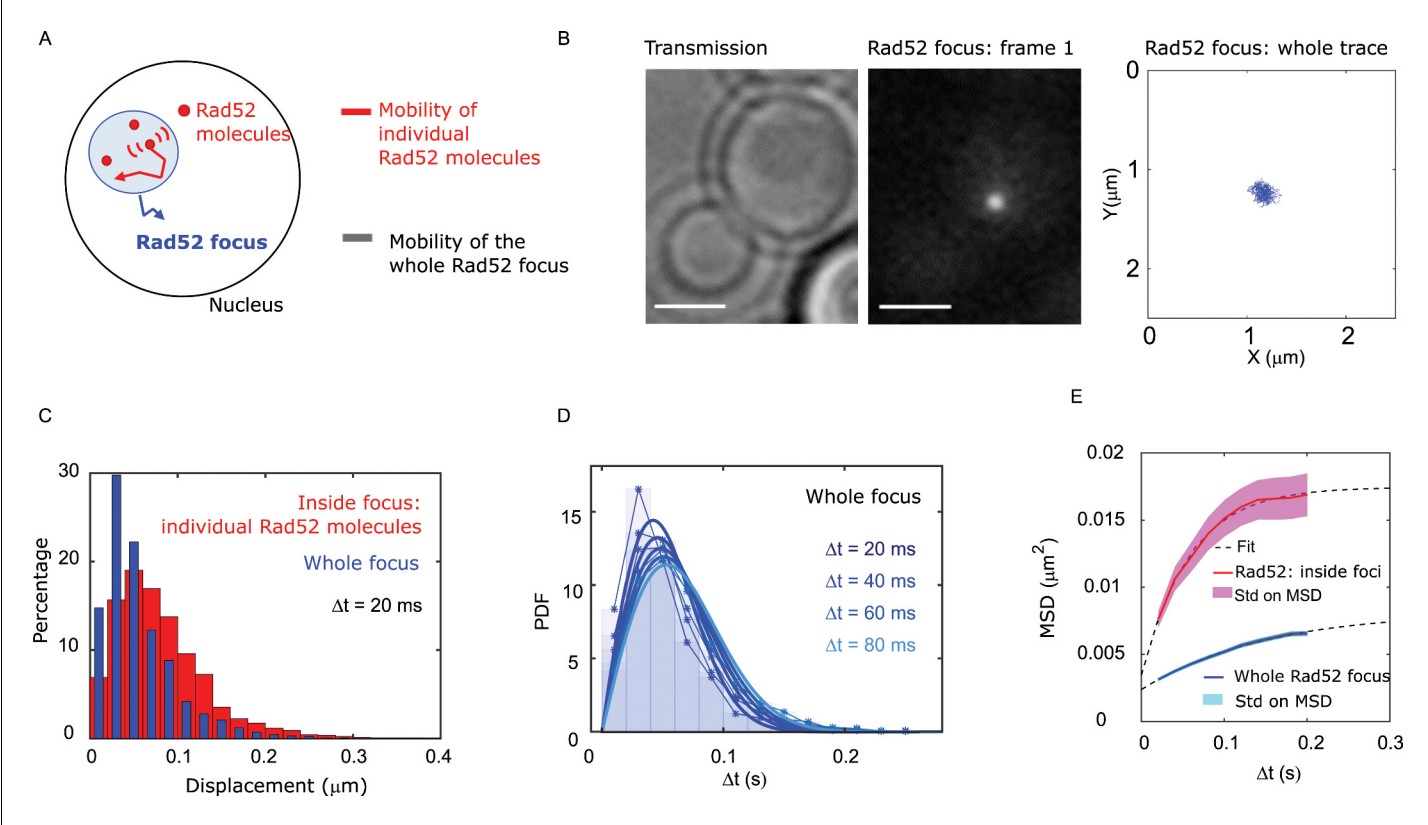

**Figure 3.** Individual Rad52 molecules are more mobile than the whole focus. (**A**) Schematic of the experiment: to shade in light the internal dynamics of Rad52, we compare the mobility of the entire Rad52 foci, shown in light red, with the mobility of individual Rad52 molecules located inside foci, in red. The mobility of the entire Rad52 focus is measured using a strain harboring Rad52-mMaple and a single I-*Sce*I DSB at *LYS2* (see Materials and methods). We used high photo-activation illumination to simultaneously activate all Rad52-mMaple and image the entire foci as a single large spot. Rad52 foci were tracked at 20 ms time-intervals in two dimensions. (**B**) Typical image of a haploid cell harboring a Rad52-mMaple focus after a 2h-induction of a single DSB. Left: transmission image; Middle: typical frame of a movie in which we see the whole focus; Right trajectory of the whole focus after analysis. The bar scale represents 1 µm. (**C**) Displacements histogram of the entire Rad52 foci (grey), compared with individual Rad52-Halo located inside foci (red, same data shown in *Figure 2F* in red). Both displacements are measured after 2 hr of galactose induction at 20 ms time intervals in 2-dimensions. F14S/G2 cells harboring a Rad52 focus are analyzed, representing 131 traces (mean length 24 frames), and 3015 translocations of 20 ms time-intervals. (**D**) PDF of the entire Rad52 foci calculated for 20, 40 and 60 ms time intervals. Plain lines represent a 1-population fit of the PDF. (**E**) MSD of the entire Rad52 foci (green) *versus* individual Rad52 molecules located inside foci (red, same data as *Figure 2G*). The MSD of individual Rad52 molecules inside foci is fitted with a confined model (dotted line) while the MSD of the entire Rad52 focus is fitted using an anomalous model (see Methods).

The online version of this article includes the following figure supplement(s) for figure 3:

**Figure supplement 1.** Nature of motion: whole Rad52 focus *versus* individual Rfa1 and Rad52 molecules inside foci (**A**): Blue: MSD of the whole Rad52 focus in S/G2 cells (14 cells); Red: fit with a model of anomalous diffusion (see Materials and methods).

## Inside foci, individual Rad52 molecules diffuse faster than the repair focus and the ssDNA-binding protein Rfa1

We next compared the mobility of individual Rad52 molecules inside foci with the mobility of the focus itself (*Figure 3A*). After inducing a single DSB for 2 hr in cells harboring Rad52-mMaple, we used high photo-activation illumination to simultaneously activate all Rad52-mMaple and image the repair focus as a single entity (*Figure 3B*). Using both displacement histograms and MSD analysis (*Figure 3C–E* and *Figure 3—figure supplement 1*), we found that Rad52 molecules diffuse 6.4 times faster than the whole focus ($D_{Rad52\ inside} = 0.032 \pm 0.006$ µm$^2$/s *versus* $D_{whole\ focus} = 0.005 \pm 0.002$ µm$^2$/s, assuming normal diffusion with a time difference of 20 ms for the whole focus). The diffusion coefficient we found is consistent with the one previously reported in response to a single DSB ($0.002 \pm 0.002$ µm$^2$/s in haploid imaged at 1 s time interval *Dion et al., 2012*). Furthermore, the shape of the MSD fits better with an anomalous diffusion of anomalous exponent $0.51 \pm 0.05$, as

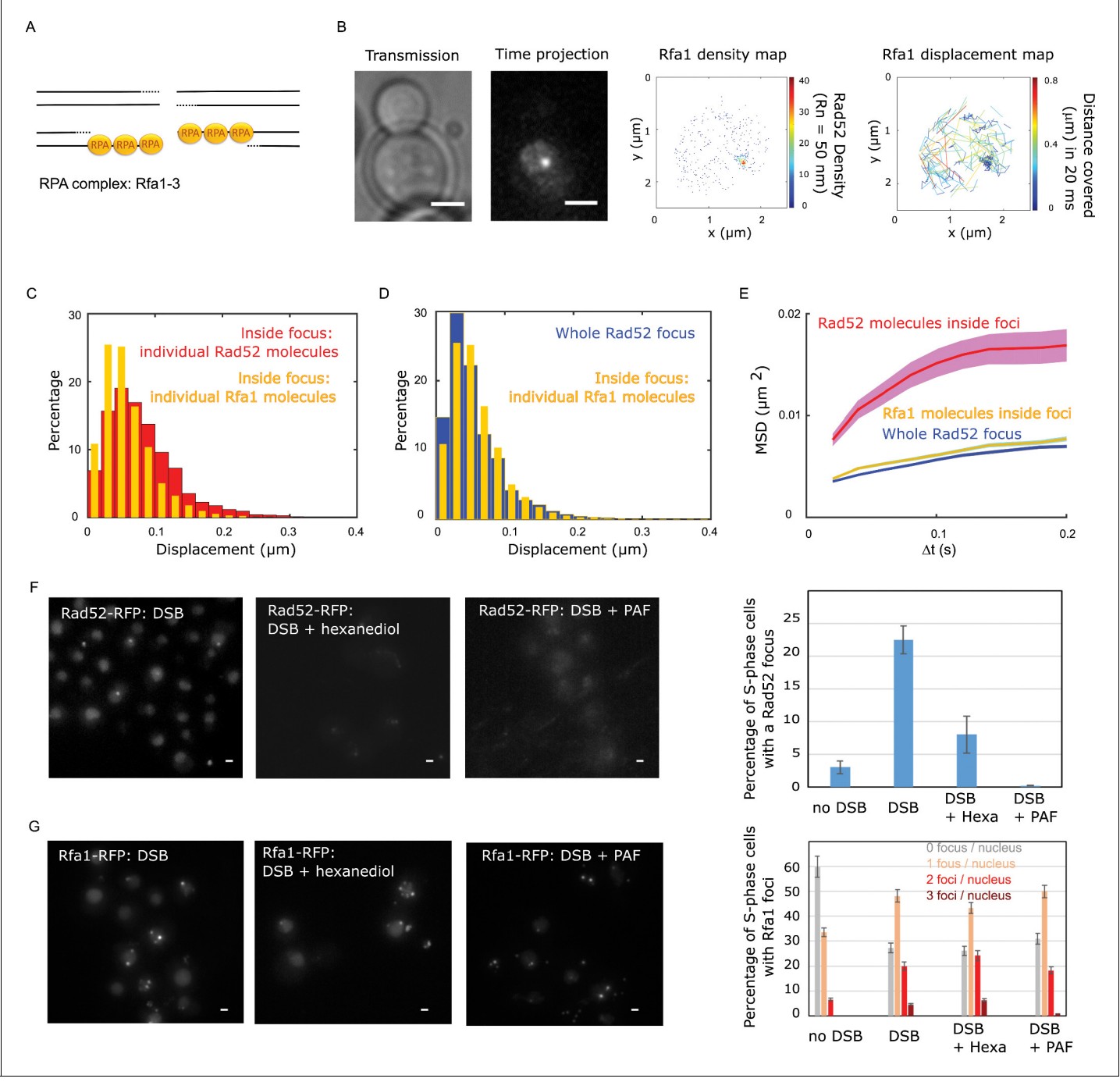

**Figure 4.** Rad52 and Rfa1 behavior strongly differs inside foci (**A**) Simplified view of a DSB repaired by HR where RPA is shown. (**B**) Typical S/G2-phase haploid cells harboring Rfa1-Halo/JF646 after 2 hr if galactose induction. From left to right: transmission image; Time-projection of the whole movie; Rfa1 density map: each spot represents a single detection of Rfa1-Halo/JF646, the color map indicates the number of Rfa1 neighbors inside a 50 nm radius disk; Rfa1 displacement map: each line represents the trajectory of a detection, the color map indicates the distance in mm covered in 20 ms. The bar scale represents 1 μm. (**C**) Displacement histograms of individual Rfa1 molecules (yellow) and individual Rad52 molecules (red) inside foci. The time interval is 20 ms. 29 S/G2 cells were analyzed, representing 621 traces (mean length of 12.4 frames) and 7095 displacement of 20 ms time intervals. (**D**) Displacement histograms of individual Rfa1 molecules inside foci (yellow) and whole Rad52 foci (grey). The time interval is 20 ms. (**E**) MSD of individual Rfa1 molecules inside foci (yellow), individual Rad52 molecules inside foci (red, same data as *Figure 3E*) and the whole Rad52 focus (green, same data as *Figure 3E*). MSDs of Rfa1 molecules and the whole Rad52 focus are fitted with an anomalous model while the MSD of individual Rad52 is fitted with a confined model. (**F**) Typical haploid cells expressing Rad52-RFP and an I-*Sce*I inducible DSB at *LYS2,* observed by wide field microscopy. First panel: a DSB is induced for 2 hr; second panel: a DSB is induced for 2 hr including 30 min with 10% hexanediol and 10 μg/ml of digitonin; third panel: a DSB is induced for 2 hr and cells are fixed with fixation with 4% paraformaldehyde or 10 min; fourth panel: quantification of the images

*Figure 4 continued on next page*

*Figure 4 continued*

presenting the percentage of S/G2 phase cells with a Rad52 focus. 200 cells are analyzed in each condition. The bar scale represents 1 µm. (G) Typical haploid cells expressing Rfa1-RFP and an I-*Sce*I inducible DSB at *LYS2,* observed by wide field microscopy. Cells are images in the same conditions as in figure (F). Since Rfa1 form several foci per nucleus, the quantification shows the number of foci per nucleus (0bar 3 foci). The bar scale represents 1 µm.

The online version of this article includes the following figure supplement(s) for figure 4:

**Figure supplement 1.** Dissolution of P-bodies with 10% hexanediol teatment.

---

previously reported for damaged chromatin at that time scale (*Miné-Hattab et al., 2017*). Thus, our results show that individual Rad52 molecules are highly mobile within the repair focus.

We then compared the mobility of individual Rad52 molecules with another component of repair foci, the Rfa1 subunit of the single-strand binding factor RPA (*Symington et al., 2014*; *Figure 4A*). Using a strain expressing Rfa1-Halo and experiencing a single I-*Sce*I DSB, we imaged individual Rfa1-Halo/JF646 2 hr after DSB induction with the same illumination conditions as Rad52-Halo (*Figure 4B*). Comparing the displacement histograms of individual Rfa1 and Rad52 molecules, we found that Rfa1 is less mobile than individual Rad52 molecules inside foci (*Figure 4C*) and exhibits a mobility similar to whole Rad52 foci (*Figure 4D*). Using both Rfa1 displacement histograms and MSD analysis inside foci, Rfa1 exhibits an apparent normal diffusion coefficient $D_{Rfa1} = 0.006 \pm 0.001$ µm$^2$/s (*Figure 4E*, assuming normal diffusion with a time difference of 20 ms). Moreover, the MSD of Rfa1 inside foci is similar to the entire focus, with an anomalous exponent of $0.56 \pm 0.05$ (*Figure 4E* and *Figure 3—figure supplement 1*).

Overall, we found that individual Rad52 diffuses approximately six times faster than the whole focus or individual Rfa1 molecules, and exhibits confined diffusion while Rfa1 and the repair focus follow anomalous diffusion.

## Rad52 high internal mobility and type of motion inside foci is conserved in diploid cells

We then verified if our observations held in diploid cells in which a repair template is available. We investigated the mobility of individual Rad52-Halo/JF646 molecules in living diploid cells harboring an I-*Sce*I cut site at one of the *LYS2* loci (see *Supplementary file 2* and *Figure 2—figure supplement 5*). Using the same illumination settings as in haploid cells, we measured the mobility of Rad52-Halo/JF646 after 2 hr of DSB induction in cells harboring a Rad52 focus. Consistent with the results obtained in haploid cells, Rad52 exhibits three distinct diffusive behaviors, the slowest population being localized in the repair focus (*Figure 2—figure supplement 5*, $D_1 = 1.06 \pm 0.05$, $D_2 = 0.22 \pm 0.02$ and $D3 = 0.033 \pm 0.002$ µm$^2$/s, p=<$10^{-7}$ 0, p=$10^{-7}$ and p=0.34, two-sided KS test for the 1-, 2- and 3-population fits respectively). The MSD analysis revealed a confined motion inside foci with a confinement radius similar to the one obtained in haploid cells (Rc $_{Haploid, inside foci}$ = 124 ± 3 nm and Rc $_{Diploid inside foci}$ = 125 ± 3 nm, *Figure 2—figure supplement 5*). Also similarly to the haploid cells, the diffusion coefficient of molecules inside foci is eight fold faster that the focus itself ($D_{diploid, inside}$ = $0.016 \pm 0.001$ µm$^2$/s, *Figure 2—figure supplement 5*, versus $D_{whole focus}$ = $0.002 \pm 0.001$ µm$^2$/s from *Miné-Hattab et al., 2017*, assuming normal diffusion with a time difference of 20 ms).

We thus conclude that the high internal dynamics of Rad52 inside foci, and the capacity of Rad52 to explore the whole focus conserved in diploid cells where DNA damage can be repaired.

## Rad52 but not Rfa1 foci are partially dissolved under 1.6-hexanediol treatment

The aliphatic alcohol hexanediol has been proposed as a tool to differentiate between liquid-like and solid-like assemblies in living cells (*Kroschwald et al., 2017*). We thus tested the effect of 1.6-hexanediol treatment both on Rad52 and Rfa1 foci formed in response to a single I-*Sce*I induced DSB. Aliphatic alcohol 1,6-hexanediol perturbs weak hydrophobic interactions. Several *in vivo* studies in yeast and mammalian cells showed that 1.6-hexanediol treatment dissolves dynamic liquid-like assemblies such as P bodies, whereas solid-like assemblies, such as protein aggregates and cytoskeletal assemblies, are largely resistant to hexanediol (*Kroschwald et al., 2017*; *Strom et al., 2017*).

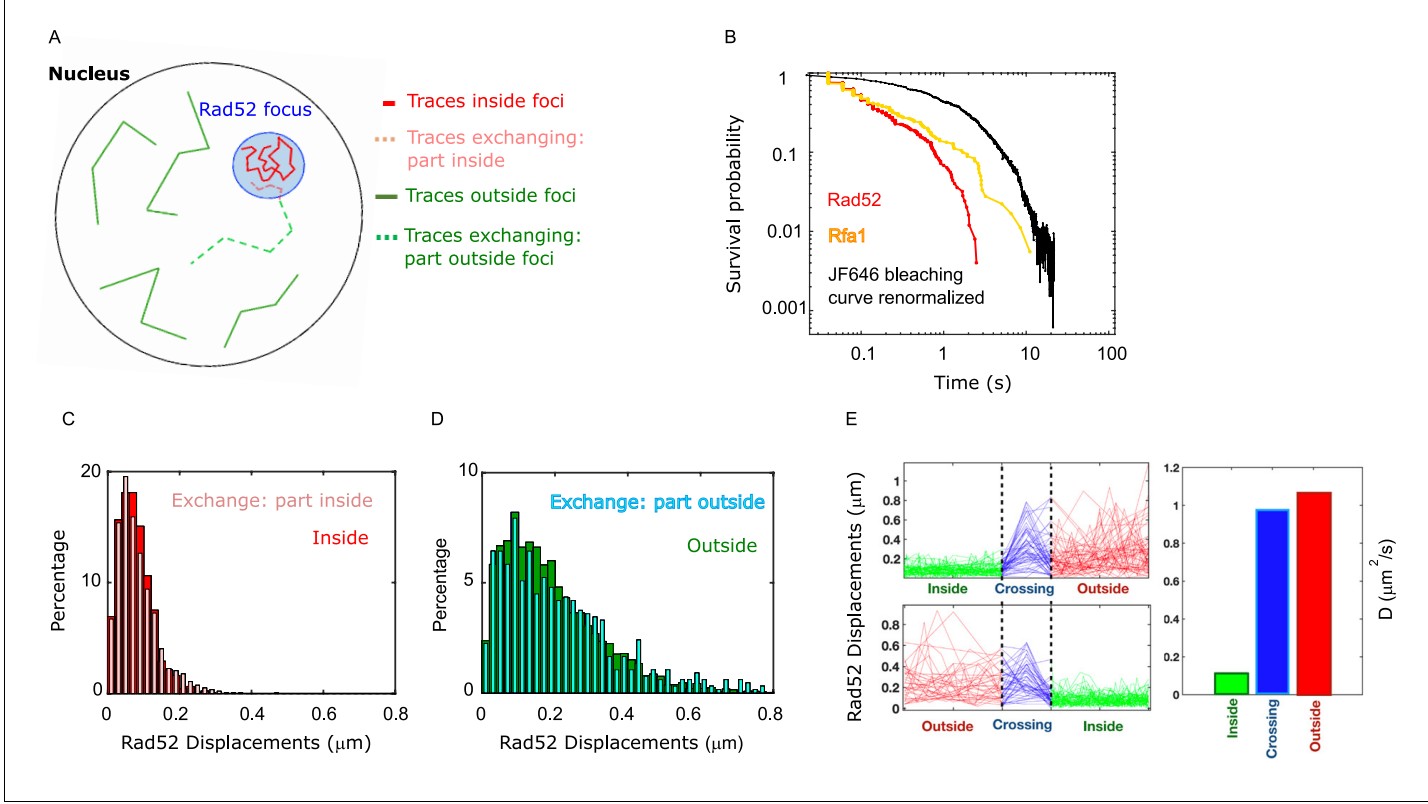

**Figure 5.** Rad52 diffusion coefficient changes when molecules enter and escape repair foci. (**A**) Illustration of the three categories of Rad52 traces observed in response to a single DSB in the nucleus: (i) traces staying inside repair foci (plain red), (ii) traces crossing foci boundaries (dotted lines), with red-dotted lines for the part inside foci and blue-dotted lines for the part outside; (iii) traces staying outside of the foci (plain blue). (**B**) Survival probability curve of Rad52 molecules inside foci (red), Rfa1 (yellow) and renormalized bleaching curve of the JF646 (black) (see *Figure 1—figure supplement 3*). (**C**) Light red: displacement histogram of traces represented as dotted red lines in *Figure 4A* (travelers, part inside foci). Dark red: displacement histogram of traces represented in plain red in *Figure 4A* (traces inside foci). (**D**) Light blue: displacement histogram of traces represented as dotted blue lines in *Figure 4A* (travelers, part outside foci). Dark blue: displacement histogram of traces represented in plain blue in *Figure 4A* (traces outside foci). (**E**) Left: Step size for traces inside the focus (green), crossing (blue), and outside the focus (red). The x-axis is squeezed so that all traces take the same space, in order to visually compare the step sizes. Above: traces starting inside and ending outside. Below: traces starting outside and ending inside the focus. Right: Bar plot showing the estimated diffusion coefficient calculated from all the traces.

The online version of this article includes the following figure supplement(s) for figure 5:

**Figure supplement 1.** Trace length distribution of Rad52 molecules inside foci in the presence of o1 DSB.

We first verified the efficiency of 1.6-hexanediol to dissolve liquid-like assemblies using a control strain expressing fluorescent P-bodies (Edc3-GFP) as described in *Kroschwald et al., 2017* (See *Figure 4—figure supplement 1* and Methods). After 30 min of treatment, the number of Edc3 foci per cell significantly decreases (*Figure 4—figure supplement 1* and Methods). We thus applied the same treatment on strains expressing Rad52-FRP or Rfa1-RFP and harboring a single I-*Sce*I cut-site, and we imaged these cells on a regular wide field microscope (*Figure 4F and G* respectively). Three conditions were analyzed: (i) cells after 2 hr of DSB induction (left panel), (ii) cells after 2 hr of DSB induction including 30 min of 1.6-hexanediol treatment (middle panel) and (iii) cells after 2 hr of DSB induction followed by 10 min of fixation with paraformaldehyde (see Methods). We found that the percentage of S-phase cells containing a Rad52 focus significantly decreases after 1.6-hexanediol treatment (22 ± 2% to 8 ± 3% respectively), consistent with recent work (*Oshidari et al., 2020*). In contrast, the percentage of Rfa1 foci remains similar with or without hexanediol treatment (*Figure 4G*, left and middle panels). We also noticed that after fixation with several classical fixation methods (Methods), Rad52 foci are not visible and Edc3 foci are significantly less abundant (*Figure 4F* and *Figure 4—figure supplement 1* respectively), while Rfa1 foci are not altered by fixation (*Figure 4G*).

In conclusion, Rad52 but not Raf1 foci show a behavior similar to P-bodies in response to 1.6-hex-anediol or paraformaldehyde treatments indicating different modes of foci formation for these two proteins, in agreement with their difference in mobility.

## Rad52 molecules change diffusion coefficient when entering or escaping repair foci

The way molecules diffuse around the boundary of membrane-less sub-compartments is crucial for defining their physical properties. We analyzed Rad52 diffusion following DSB in more detail (*Figure 5A*). First, we estimated the residence time of Rad52 molecules inside foci by calculating the survival probability of Rad52 molecules in a focus. Importantly, by comparing this curve with the bleaching time of the JF646, we checked that JF646 bleaching is not a limiting factor (*Figure 5B*). We found that the mean residence time of Rad52 inside foci, defined as the integral of the survival probability curve, is ~240 ms while Rfa1 molecules stay longer (560 ms) (*Figure 5B*).

So far, we compared Rad52 diffusion inside *versus* outside repair foci; however, some trajectories cross the focus boundary, and those were separated in two parts. To understand how Rad52 diffuses around foci boundary, we first quantified the different categories of traces: (i) traces staying inside repair foci during the time of the acquisition, (ii) traces crossing focus boundaries and (iii) traces stay-ing outside foci (*Figure 5A*). Among the first 2 categories of traces, 70% cross a focus boundary at least once during the 20 s movie, while 30% stay inside foci, indicating that Rad52 foci are in constant exchange with the rest of the nucleus. As a comparison, only 33% of Rfa1 molecules cross a focus boundary. Of course, these exchange rates are specific to our acquisition settings.

We then focused only on Rad52 trajectories crossing repair focus boundaries, which we call 'travelers' (dotted lines in *Figure 5A*). To test whether Rad52 molecules change their diffusive behavior when entering or escaping repair foci, we calculated the displacement histograms of these traces, distinguishing portions inside and outside foci (*Figure 5A*, dotted red and dotted blue traces respectively). We observed that Rad52 molecules change diffusive behavior sharply when crossing the focus boundary. Molecules crossing into foci (category 2) diffuse similarly to molecules staying inside foci, as long as they are in the focus (*Figure 5C*); when crossing the boundary outward, they start diffusing like molecules outside of foci (*Figure 5D*). To have a visual representation of this change in diffusion, we displayed the norm of the displacement vectors over time of all travelers, centering all the traces when they cross the boundary (*Figure 5E*). In other words, Rad52 molecules clearly change diffusion coefficient when entering or escaping repair foci, adopting the diffusive behavior of the surrounding Rad52 molecules in their environment.

## Following multiple DSBs, Rad52 confinement scales with focus size

Following the induction of multiple DSBs, it has been shown in yeast, Drosophila and human cells that repair sites move and collapse into larger units, or 'clusters' (*Aten et al., 2004*; *Aymard et al., 2017*; *Chiolo et al., 2011*; *Krawczyk et al., 2012*; *Lisby et al., 2003*; *Neumaier et al., 2012*; *Oshidari et al., 2020*; *Schrank et al., 2018*; *Waterman et al., 2019*), presumably to facilitate DSB repair progression by increasing the local concentration of repair proteins. However, the structure of collapsed DSBs remains unknown. We thus used PALM to investigate the structure of Rad52 foci formed in response to multiple DSBs at the single molecule level. We compared haploid strains har-boring a single I-*Sce*I site (*Figure 2A*) with haploid strains harboring 2 I-*Sce*I sites on different chro-mosomes (*Figure 6A* and Methods). In both cases, DSB(s) were induced for 2 hr and we observed a single focus per nucleus in most cells, as previously described (*Lisby et al., 2003*). First, we mea-sured the mobility of the entire Rad52 focus induced by a single DSB *versus* 2 DSBs, at 20 ms time intervals. Using strains expressing Rad52-mMaple with high photo-activation, we found that both foci exhibit similar mobility (*Figure 6B*). This is consistent with previous studies reporting no change in chromatin diffusion coefficients when inducing more DSBs despite a dramatic increase in the chro-matin confinement radius (*Miné-Hattab and Rothstein, 2012*).

We then compared the size, number of Rad52 molecules and density of foci formed in response to 1 *versus* 2 DSBs (*Figure 6C*). Since the focus size is under the diffraction limit, we used PALM in haploid cells expressing Rad52-mMaple. Since we observed that Rad52 foci are not visible after fixa-tion with classical fixation methods, we performed PALM in living cells (*Izeddin et al., 2011*) (refer-eed as live PALM, see Methods). We found that foci induced by 2 DSBs are larger than those

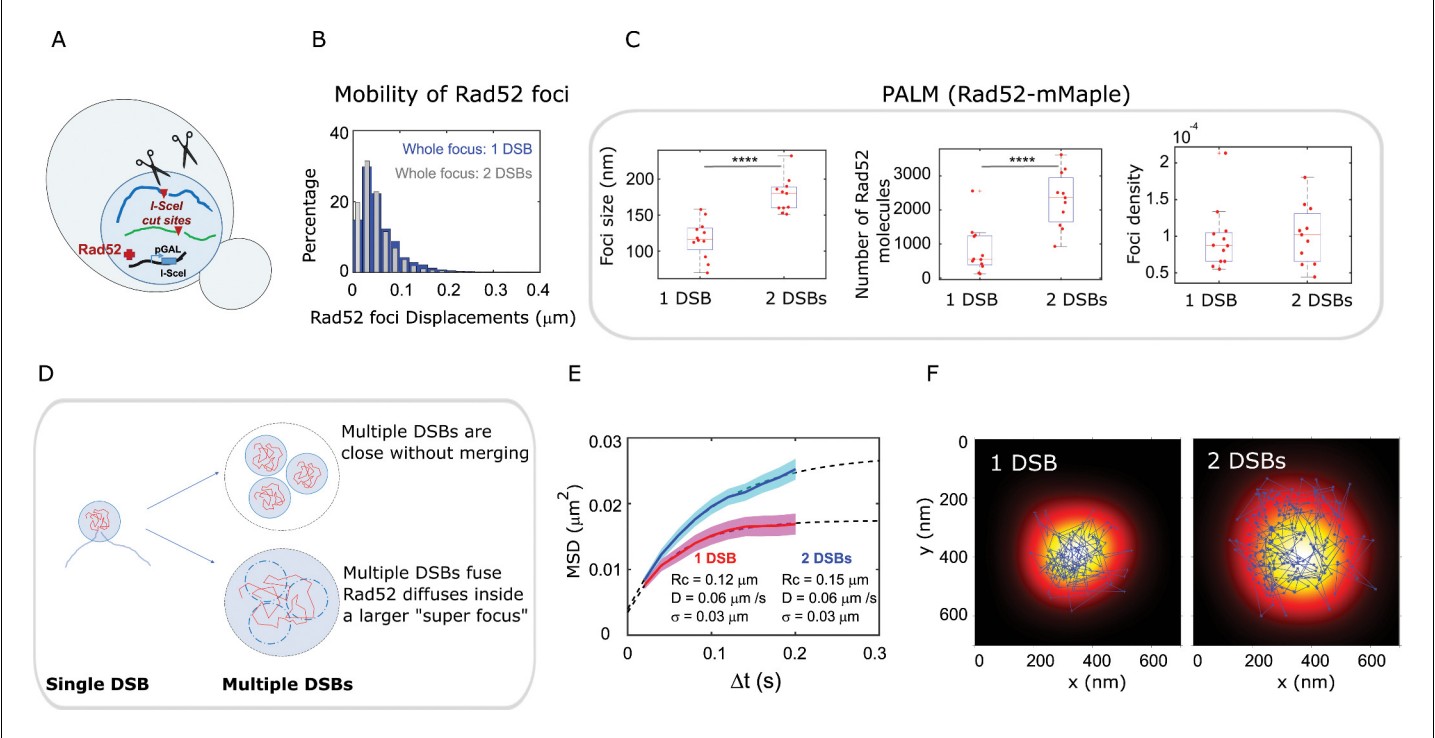

**Figure 6.** Rad52 foci in the presence of multiple DSBs. (**A**) Design of the experiment. We induced 2 I-SceI DSBs (at *LYS2* and *HIS3*) in haploid cells harboring Rad52-Halo (see Methods). The DSB is induced for 2 hr and fluorogenic JF646 dyes are added to the medium during the last hour of incubation prior visualization by SPT. Individual Rad52-Halo/JF646 are tracked at 20 ms time intervals (50 Hz), during 1000 frames. Only cells harboring a Rad52 focus are analyzed. (**B**) Displacement histogram whole Rad52 foci: Blue: after the induction of a single DSB (same data as 3C); Red: after the induction of 2 DSBs. Rad52 foci were tracked at 20 ms time intervals after 2 hr of galactose induction. For the experiments following the induction of 2 DSBs, we examined 14 cells, representing 702 traces (mean length of 9.8 frames) and 6220 displacements of 20 ms time intervals. (**C**) Top left: Rad52 Focus size measured for 1 DSB-induced foci *versus* 2 DSBs-induced foci. We performed live PALM on cells harboring Rad52-mMaple strains (see Methods). Bottom left: Estimation of the number of Rad52 molecules inside 1 *versus* 2 DSBs-induced foci, measured by live PALM. Right: Rad52 Foci density of 1 *versus* 2 DSBs-induced foci. (**D**) Illustration of the 'Cluster' and the 'Fusion' scenarii in the case of multiple DSBs. (**E**) MSD of 1 *versus* 2 DSBs-induced foci (red and blue curves respectively). Dotted lines represented a fit of the experimental MSDs, using a model of confined diffusion. (**F**) Typical example of a Rad52 trajectory represented in blue. The whole focus is shown in the background using a Gaussian blur of each Rad52 detections contained in this focus. Left: 1 DSB-induced focus; right: 2 DSBs-induced focus.

induced by a single DSB, ($170 \pm 20$ nm *versus* $116 \pm 20$ nm respectively, $p=1\times10^{-4}$, Wilcoxon-Mann-Whitney test). In addition, the number of Rad52 molecules forming foci varies from $880 \pm 600$ for 1 DSB induced foci to $2300 \pm 811$ for 2-DSBs foci ($p=8\times10^{-4}$, Wilcoxon-Mann-Whitney test). Thus, focus density is indistinguishable between foci induced by 1 *versus* 2 DSBs (*Figure 6C*, right panel, $p=0.73$, Wilcoxon-Mann-Whitney test).

We propose 2 views of DSBs clustering illustrated in *Figure 6D*. In the first view, multiple DSBs cluster and stay close without merging (top figure). In the second, multiple DSBs fuse together and molecules are shared between the different DSBs (bottom figure). In the latter case, different foci merge together giving rise to a larger focus where Rad52 molecules explore the entire space; thus the radius of confinement should scale with the focus size. In the former case, Rad52 molecule should have the same confinement radius as in the situation of single DSB if the clustered foci remain sufficiently distant. If the two foci are sufficiently close for Rad52 molecules to 'jump' from a focus to another one, we expect to observe bimodal trajectories, exhibiting confined diffusion in the first focus, followed by one or several larger steps and confined diffusion in a second focus. Since the average residence time inside a focus induced by 1 DSB is 240 ms (12 frames) and traces are often longer than that inside foci (*Figure 5—figure supplement 1*), such traces should be easily observable if present. Here, we found that upon 1 DSB, Rad52 diffuse inside foci with smaller confinement radius ($124 \pm 3$ nm) compared to foci induced by 2 DSBs ($156 \pm 3$ nm), supporting the second view where foci fuse (*Figure 6E*). Importantly, this confinement radius found by SPT is very close to the

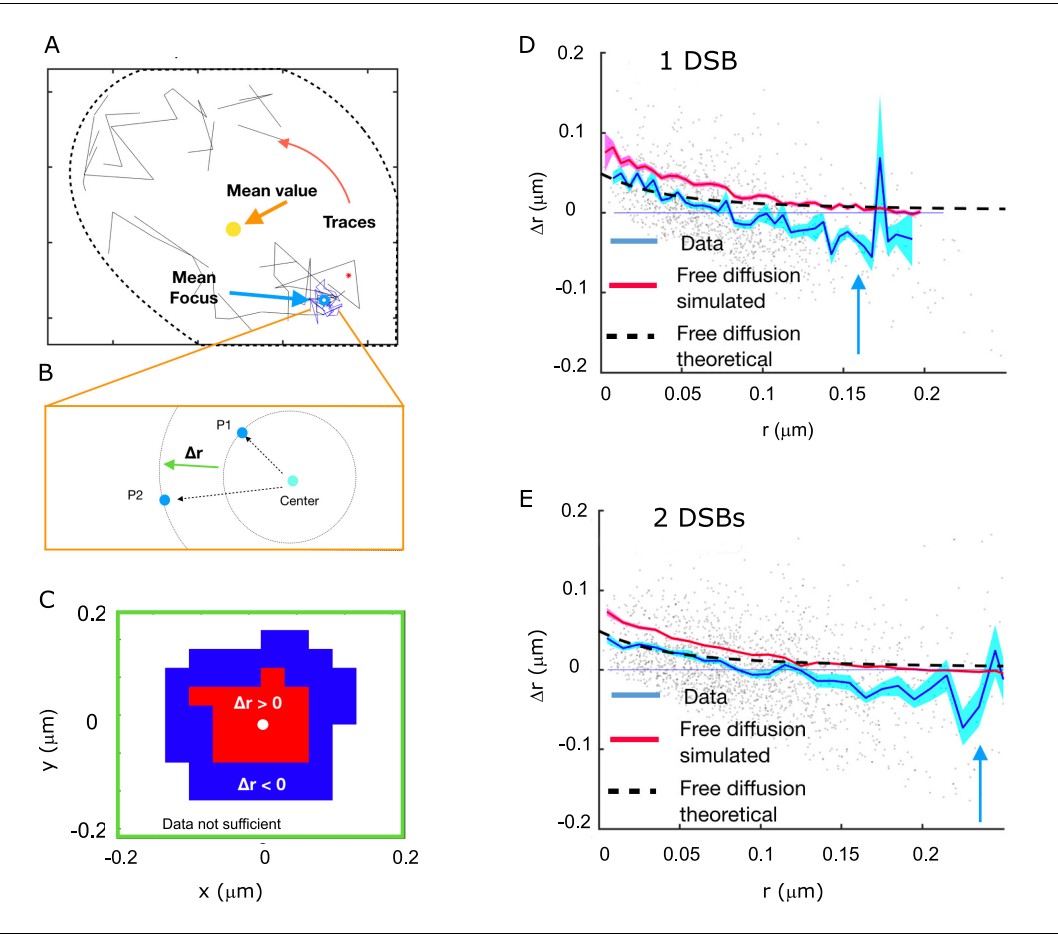

**Figure 7.** Rad52 shows attractive motion around the focus. (**A**) Representative cell, with traces outside the focus (black) and inside the focus (blue). Cell nucleus is estimated from the traces. (**B**) Schematic showing the definition of the radial displacement Δr, defined by two subsequent data-points. (**C**) Sign of the average radial displacement as a function of particle position relative to the focus, averaged over all cells. (**D**) Radial displacement Δr versus r (distance to the center of the focus prior to displacement) over all cells with 1 DSB. Cyan line indicates the mean value. Black line corresponds to the expected movement for free diffusion with diffusion coefficient equal the one inside the focus (assuming the center of focus is exactly known). Red line corresponds to a free diffusion, simulated with same trace length and the same diffusion coefficient as the data traces and the same method for estimating the center of the focus. (**E**) Same as (**D**) but for cells with 2 DSBs.

mean size of Rad52 foci previously obtained by live PALM. The correlation between focus size and confinement radius indicates that Rad52 is able to explore the entire focus space. To have a visual representation of Rad52 exploration inside foci, we overlaid a reconstruction of a focus with one of the longest Rad52 trajectories contained inside that focus. The reconstruction of the whole focus is obtained by taking all the detections inside it, except the ones belonging to the overlaid trajectory. These detections are represented by a blurred spot using a Gaussian smoothing kernel around each detection. Two typical foci are shown in this representation: a focus formed in response to a single DSB (*Figure 6F*, left panel) *versus* a focus induced by 2 DSBs (*Figure 6F*, right panel). This representation clearly shows that Rad52 explores the focus and that its confinement radius scales with the focus size. In addition, using this representation, we never observe bimodal trajectories, allowing us to rule out clustered but non-merged foci.

## Rad52 shows attractive motion around the focus

Next, we wanted to estimate whether there was a region of attraction for the molecules of Rad52 inside the focus. For each cell, we considered a set of traces, of which a subset was defined to be

inside the focus (**Figure 7A**). We quantified the radial movement of molecules relative to the center of the focus in terms of the change in the distance to that center between two subsequent time steps, which we denote Δr (**Figure 7B**). To obtain this measure we needed to estimate the position of the center of the focus. Since we observed that the focus itself was diffusing ~6 times more slowly than the Rad52 molecules, we approximated the focus as the average particle position in each analyzed trace (see Methods). Concentrating on molecules close to the boundary, we identified a region of attraction with an average movement of molecules towards the center of the focus (**Figure 7C**).

To get a more detailed map of displacements, we plotted the radial movements of all particles as a function of their original position relative to the center for both 1 and 2 DSB's (**Figure 7E and F**). The mean radial movement, outside the focus but close to its boundary, has negative values, which is consistent with an attractive force towards the center of the focus. This does not mean that this potential attracts molecules to the center of the focus, but it hinders the escape from the focus region. Inspired by the observation that molecules are attracted to the center of the focus, we computed the difference in the free energy. Using the estimated fraction of molecules belonging to the slow diffusion population (**Figure 2**), we extracted a change in the Gibbs free energy difference $U_0$ between the inside of the focus and the surroundings. This was done by estimating the increased density of molecules inside the focus using Boltzmann's law for the fraction of slowly diffusing molecules, $p = \frac{e^{-U_0/k_B T} V_F}{V_0 - V_F + e^{-U_0/k_B T} V_F}$, where $V_F$ is the volume of the focus, as calculated by the MSD, $V_0$ is the volume of the observable frame. Inverting this relation for $U_0$ and injecting the measured values of $p$, $V_0$, and $V_F$ yields $\frac{U_0}{k_B T} \approx -5.5$ (**Figure 7G**). However, detection noise (as estimated in **Figure 2**) and low statistics far away from the focus make it difficult distinguish between a continuous potential and a surface potential.

## Discussion

The nucleus contains membrane-less sub-compartments that form and disassemble according to the needs of the cell at times scales relevant for their biological function. Repair foci provide a powerful example for studying such sub-compartments because it is possible to induce their formation at will *in vivo* and compare the behavior of repair proteins before and after DSB induction. Following DSB, HR proteins relocalize from a diffuse nuclear distribution to a sub-nuclear focus at the DNA damaged site (**Lisby et al., 2004**). To understand this process, here we observed for the first time how Rad52 molecules diffuse inside living nuclei in the presence and in the absence of DSB using single molecule microscopy.

### Dynamic behavior of individual Rad52 in living cells

Tracking single molecules of the repair factor Rad52 at 50 Hz, we observed that in the absence of DSB, Rad52 exhibits two distinct diffusive behaviors characterized by apparent diffusion coefficients $D_1 = 1.15$ μm$^2$/s, and $D_2 = 0.27$ μm$^2$/s with a ratio$_{2/1}$ = 2/3 (see **Supplementary file 1**). These two populations were observed both in G1 and S phase suggesting that the slower population does not correspond to transient binding of Rad52 at replication forks. Assuming Brownian diffusion, a diffusion coefficient of 1.15 μm$^2$/s implies that Rad52 can go across a yeast nucleus in as little as 700 ms. Moreover, tracking free Halo-NLS, we estimate the dynamic viscosity inside yeast nuclei to 0.1 Pa.s, compared to 10$^{-3}$ Pa.s for water at 20°C, 0.013 P.s in Hela cells nuclei (**Liang et al., 2009**) and 6 Pa.s for honey. Using the Stokes-Einstein equation, we showed that the two diffusion coefficients measured for Rad52 agree with the ones predicted for the monomeric and multimeric forms of Rad52 observed *in vitro* (**Saotome et al., 2018**; **Shinohara et al., 1998**). Thus, single molecule tracking of Rad52 suggests that both forms co-exist in living nuclei, two thirds of them being multimers.

In response to a DSB, Rad52 accumulates at the damaged site forming highly concentrated foci (Lisby et al., n.d.). The presence of a DSB does not influence the mobility of Rad52 outside of the focus, consistent with (**Essers et al., 2002**), but inside foci, Rad52 exhibits an intermediate diffusive behavior, slower than free Rad52 but faster than damaged DNA ($D_{Rad52,inside}$ = 0.032 ± 0.006 μm$^2$/s). Taking the measured diffusion coefficient and confinement radius, Rad52 molecules inside foci start feeling the effect of its boundary at $t_c = R_c^2/(6 \cdot D) = 80 \pm 3$ ms, representing four time-steps of 20 ms (see Materials and methods and [**Klein et al., 2019**]). Assuming a multimeric form inside foci, the Rad52 diffusion coefficient measured here would be consistent with a viscosity inside the focus

of 1 ± 0.1 Pa.s, and it would reach 2.4 ± 0.1 Pa.s assuming a monomeric form. These viscosities are respectively 8 to 20 times higher than the nucleus viscosity estimated before damage ($\nu_{\text{yeast nucleus}}$ = 0.122 ± 0.004 Pa.s). In human cells (U2OS), a similar viscosity inside repair foci has been found (2.5 Pa.s) by measuring the diffusion of 53BP1 molecules within foci (*Pessina et al., 2019*). Although human nuclei are ~25 times less viscous than yeast, the viscosity inside repair foci appears similar in yeast and humans.

## Different dynamics for Rad52 and Rfa1 within repair foci

Comparing the dynamics of Rad52 with the single strand binding protein Rfa1, we observed a very different behavior in response to a single DSB. Inside foci, individual Rfa1 molecules move ~6 times slower than Rad52 molecules. Furthermore, individual molecules of Rfa1 and Rad52 exhibit a different type of motion (*Figure 4* and *Figure 3—figure supplement 1*). Rfa1 molecules inside foci follow anomalous diffusion with an anomalous exponent ($\alpha_{\text{Rfa1}}$ = 0.56 ± 0.05) consistent with the Rouse model, a behavior similar to the focus itself ($\alpha_{\text{focus}}$0.51 ± 0.05) or to damaged chromatin when imaged at the same frequency (50 Hz) (*Miné-Hattab et al., 2017*). In contrast, individual Rad52 molecules are highly mobile and exhibit confined motion inside the focus (*Figure 4* and *Figure 3—figure supplement 1*). Such differences in diffusion coefficient and type of motion between Rad52 and Rfa1 indicate that most of Rad52 molecules are not bound to the Rfa1-coated ssDNA inside foci, in agreement with (*Essers et al., 2002*). In addition to high internal dynamics, Rad52 exhibits a higher rate of exchange than Rfa1with the surrounding nucleoplasm. The rapid exchange rate observed here for individual Rad52 molecules is in good agreement with FLIP (fluorescence loss in photobleaching) experiments in yeast cells (*Oshidari et al., 2020*). Fluorescence Recovery After Photobleaching (FRAP) experiments in mammalian cells also revealed different exchange rates and residence times depending on the repair proteins: Rad54 and Rad52 inside foci rapidly exchange with the rest of the nucleus while Rad51 stays immobilized at the site of DNA damage (*Essers et al., 2002*).

The intermediate mobility of Rad52 molecules inside foci (faster than chromatin or Rfa1 molecules but slower than free molecules) could in principle be explained by rapid equilibrium binding of Rad52 to its target, which would effectively decrease its mobility. However, in that case we should be able to capture with SPT some intervals of time where Rad52 is bound. During these bound intervals, Rad52 should diffuse slowly with the same diffusion coefficient as Rfa1 and such events would lead to a strongly multi-modal distribution of displacements inside foci. We do not observe such multi-modal distribution inside foci in our data at 20 ms resolution. To be consistent with the smoothness of our displacement histogram, the binding specificity would have to be very weak, $K_d = 1/(4\,D\,a\,\tau) \gg 2$ μM (*Berg and Purcell, 1977*), where a ≈ 10 nm is the linear size of the binding site, D ≈ 1 μm²/s, and τ is the mean bound time ≪ 20 ms, conservatively assuming diffusion-limited binding. This dissociation constant is 20 to 400 times larger than the values observed *in vitro* (*Saotome et al., 2018*). Assuming a realistic range of binding dissociation constants $K_d$ ~1 nM to 1 μM, the mean bound time of $\tau = 1/(K_d\,D\,a)$ would range between 40 ms to 40 s. These values are all larger than our resolution and should be easily observable if we had a binding and free diffusion inside foci. Altogether, our data and analysis rather suggest that most Rad52 freely diffuse inside foci and only a very small fraction of Rad52 molecules are bound to the Rfa1-coated ssDNA. Since with single molecule experiments, we can follow only a fraction of the molecules, it is very unlikely to catch the few bound molecules in our assay.

## Rad52, but not Rfa1, behavior inside foci is consistent with LLPS model

Different models have been proposed to account for the formation of membrane-less compartments. The simplest model is the binding model where molecules bind and unbind their target sites (i.e. ssDNA for Rfa1). In this case, the local concentration of proteins reflects the number of binding sites, the amount of proteins available and their affinity for their binding sites, without phase separation. If the binding sites are present on chromatin or DNA, these binding sites can be close to each other increasing their local density. In a second scenario, some binding proteins can form bridges between different chromatin loci by creating loops or by stabilizing interactions between distant loci along the chromatin fiber. These interactions can be driven by interactions between chromatin binding proteins or chromatin components. In this model, referred to as 'Bridging' Model, or Polymer

Polymer Phase Separation (PPPS) model (*Erdel and Rippe, 2018*; *Miné-Hattab and Taddei, 2019*), the existence of sub-compartments relies on both the binding and bridging properties of these proteins to chromatin. A third scenario is the Liquid-Liquid Phase Separation (LLPS) Model, also referred to as 'Droplet Model' (*Hyman et al., 2014*). Unlike the binding and the PPPS models, in LLPS, proteins self-organize into liquid-like droplets that grow around a nucleation site allowing certain molecules to become concentrated while excluding others. In this framework, one should distinguish proteins able to initiate a liquid phase separation on their own (scaffold proteins) from proteins concentrating and diffusing freely within this droplet without being responsible for its formation (client proteins) (*Banani et al., 2017*).

Despite a large literature in the field, many *in vivo* tests commonly used to probe the nature of sub-compartments are extremely phenomenological and insufficient to rule out other possible mechanisms (*McSwiggen et al., 2019b*; *Miné-Hattab and Taddei, 2019*). In addition, most of the optical methods used to discriminate models are at the limit of the diffraction and suffer from severe artefacts (*McSwiggen et al., 2019a*; *Miné-Hattab and Taddei, 2019*). Overall, there is a lack of solid experimental criteria to confirm one model *versus* another in living cells, in particular at the microscopic level.

Our results reveal that Rad52, but not Rfa1 motion shares several properties of LLPS models which are conserved from haploid to diploid cells:

- First, in the LLPS model, even in the presence of a nucleation core, the large majority of proteins are not bound to chromatin (*Erdel and Rippe, 2018*; *Miné-Hattab and Taddei, 2019*). Consistent with the LLPS model, Rad52 molecules inside foci exhibit high mobility compared to the focus itself, and the statistics of confined diffusion with a confinement radius of the same size as the whole focus. In contrast with Rad52, the motion of Rfa1 molecules, which coat ssDNA, have similar diffusion coefficient and type of motion as the focus.
- Second, LLPS are characterized by a change of motion upon entering or escaping the sub-compartments (*Banani et al., 2017*; *McSwiggen et al., 2019a*). Here, we observed a sharp change in the diffusion coefficient of Rad52 molecules crossing foci boundary.
- Third, in the LLPS model, it is energetically more favorable for molecules to stay inside the sub-compartment than to leave because of the presence of an energetic potential maintaining the sub-compartment. Here, we observed the existence of an attractive potential maintaining Rad52 molecules inside the focus even if molecules are not physically bound to a polymer substrate. However, given the noise and the variable size of foci, we cannot currently describe the shape of the potential.
- Fourth, an important hallmark of LLPS is their ability to fuse, two droplets for radius R leading to a bigger droplet of doubled volume. Fusion of repair foci has been previously observed in yeast (*Oshidari et al., 2020*; *Waterman et al., 2019*) and in human cells (*Kilic et al., 2019*; *Pessina et al., 2019*; *Schrank et al., 2018*; *Sollazzo et al., 2018*) by tracking repair foci at larger time scales (several minutes). Here, using live PALM, we further show that foci resulting from 2 DSBs are 1.99 ± 0.2 larger in volume than those formed by a single DSB (156 ± 3 nm *versus* 124 ± 3 nm respectively), and present similar Rad52 densities. Moreover, focus sizes measured by PALM agree with the confinement radius of individual Rad52 molecules inside foci, showing the ability of Rad52 to explore the whole sub-compartment. Thus, our results indicate that upon 2 DSBs, Rad52 foci do not merely cluster, but form a focus of size consistent with the fusion of two foci inside which molecules explore the entire larger sub-compartment.
- Fifth, at the macroscopic level, aliphatic alcohol hexanediol, a component proposed as a tool to differentiate liquid-like from solid-like assemblies, partially dissolves Rad52 foci and P-bodies while Rfa1 foci remains as abundant. Of note, Rad52 foci as well as P-bodies (Edc3 foci) are not visible after fixation (see Materials and methods), whereas Rfa1 foci resist to the same fixation treatment, suggesting that paraformaldehyde might not be able to fix LLPS.

Other predictions of the LLPS *versus* PPPS properties have been proposed in the literature, in particular when over-expressing a protein able to form a nuclear sub-compartment. In a simple LLPS model where a sub-compartment is formed by a single species, increasing protein amounts should increase the focus size while concentrations remain the same inside foci and in the nucleoplasm (*Erdel and Rippe, 2018*; *Miné-Hattab and Taddei, 2019*). When a LLPS is formed but the observed molecule is not the main species driving the LLPS (referred as a 'client' of a LLPS), the concentration inside foci and in the background increases linearly with over-expression of the observed molecule (*Hyman et al., 2014*). Here, we found that upon different levels of Rad52 over-expression, the

background concentration increases as predicted for a 'client' (see Appendix 1 and *Appendix 1- figure 1*) suggesting that Rad52 might not be the main driving molecule responsible for the LLPS formed at the damaged site. This is in agreement with the observation that several DSBs coalesce and dissociate independently of Rad52 (*Waterman et al., 2019*). However, a recent study pointed out that LLPS composed by multiple proteins with heterotypic multicomponent interactions do not exhibit a fixed saturation concentration (*Riback et al., 2020*). Since sub-compartments are formed by a complex mixture of proteins, it is possible that the formation of a LLPS is not driven by a single species defined as a driver, but rather by the combination of several interacting proteins.

Recent studies across different model systems proposed that repair foci are LLPS (*Altmeyer et al., 2015*; *Kilic et al., 2019*; *Oshidari et al., 2020*). In human cells, it has been suggested that DSB-induced transcriptional promoters drive RNA synthesis and stimulate phase separation of repair proteins (*Pessina et al., 2019*). As mentioned in *McSwiggen et al., 2019b*, some of the criteria used to define a LLPS are at the limit of the optical resolution or are not sufficient to distinguish between a LLPS and a PPPS. For example, sub-nuclear compartments can phase separate by PPPS and collapse of such foci is sometimes wrongly interpreted as a fusion of liquid droplets (*Erdel et al., 2020*; *McSwiggen et al., 2019a*). Here, by quantifying the physical properties of Rad52 foci at the single molecule level, our study confirms that Rad52 diffusion is consistent with a LLPS at damaged sites, although Rad52 might not be the main driver of droplet formation.

Finally, we found that SUMOylation modulates the physical nature of Rad52 foci. First, the mobility of SUMOylated Rad52 molecules inside foci remains confined but becomes significantly slower than wild type Rad52 ($D_{Rad52-Sumo,\ inside}$ = 0.009 ± 0.002 µm$^2$/s compared to $D_{Rad52\ inside}$ = 0.032 ± 0.006 µm$^2$/s). Second, SUMOylated Rad52 foci are 3.4 times smaller in volume but exhibit similar intensity than wild type foci, indicating that they are denser. We propose that when all Rad52 molecules are SUMOylated, Rad52 foci transition to a gel-like sub-compartment where most of the molecules are still free to explore the whole focus but much slower than in wild type cells. Since SUMOylated Rad52 foci are smaller, it is possible that the formation of LLPS around damage sites is a mechanism to increase the pull of molecules at the DSB.

## Working model for Rad52 foci

Summing up all our observations, we propose the following model. In response to a DSB, Rad52 diffusion is not altered in the nucleus, thus Rad52 molecules likely do not exhibit a strong collective or directive motion toward the damaged DNA site. They rather arrive by simple diffusion at the DSB where they bind ssDNA coated by the RPA complex. The focus is then formed by a small seed of Rad52 molecules bound to RPA coated ssDNA, around which a large cloud of Rad52 rapidly diffuse exploring the whole focus. Similar to LLPS, Rad52 foci are dense but surprisingly dynamic: Rad52 molecules inside foci are ~6 times more mobile than ssDNA-binding proteins Rfa1 and exhibit permanent exchange with the rest of the nucleus. When escaping foci, Rad52 molecules change their diffusion behavior outside foci, suggesting that their diffusive behavior is defined by its environment. Rad52 molecules are retained inside the focus by a potential. The existence of such attractive potential supports a model where DNA is dispensable to maintain the phase separation once a given saturating concentration of multivalent binder molecules is reached. Overall, our results at the single molecule level reveal that Rad52 foci exhibit many features of LLPS at the microscopic level, consistent with recent studies using macroscopic criteria both in yeast and in human cells (*Altmeyer et al., 2015*; *Kilic et al., 2019*; *Oshidari et al., 2020*; *Pessina et al., 2019*). Our results provide a quantitative measurement of Rad52 behavior once the focus is formed and future studies will be necessary to investigate Rad52 foci formation.

In the future, single molecule microscopy combined with genetics will be a powerful method to shed light on the role of mutants affecting diffusion, residence time, focus size and the type of motion. In particular, further studies in different genetic backgrounds might reveal which exact mechanisms drive the formation and the physical nature of Rad52 foci. More generally, applying single molecule microscopy to other biological processes may reveal that nuclear sub-compartments with very similar aspects and sharing some macroscopic features of LLPS are in fact formed by other alternative mechanisms. For example, silencing sub-compartments resemble Rad52 foci: they fuse and disassemble in specific metabolic conditions. However, single molecule microscopy reveals that Sir3 proteins diffuse very differently than Rad52 (our unpublished results) and do not behave as expected in a LLPS (*Miné-Hattab and Taddei, 2019*). In human cells, Herpes Simplex Virus appears

to share many properties commonly attributed to LLPS, but SPT measurements have recently shown that they are formed through a distinct compartmentalization mechanism (*McSwiggen et al., 2019b*). Similarly, although several studies proposed that heterochromatin sub-compartments are LLPS, recent results question a simple LLPS mechanism and are consistent with a collapsed chromatin-globule model (*Erdel et al., 2020*). As the list of proteins that can undergo phase separation is growing in the literature (*Mészáros et al., 2019*; *You et al., 2020*), it will be essential to confirm the nature of sub-compartments using *in vivo* single molecule approaches. As we obtain higher-resolution biophysical insight, what appear to be droplets at first glance might in fact follow more complex models. Therefore it is becoming necessary to define solid criteria and observables at the microscopic level to distinguish between different models and to develop alternative models of membrane-less sub-compartments.

## Materials and methods

All strains used in this work are isogenic to RAD5+ W303 derivatives (*Supplementary file 2*).

### Strain constructions

Strain harboring Rad52-mMaple (*McEvoy et al., 2012*) or Rad52-Halo are constructed using crisper-Cas9 technic: the fusion is made at the Rad52 N-terminus with a 48 bases linker (CGTACGC TGCAGGTCGACGGAGCAGGTGCTGGTGCTGGTGCTGGAGCA) or a 15 bases linker (AG TGGAGGCGGAGGT), respectively. For strains harboring Rad52-mMaple, we first built plasmid pAT475 containing the cas9 sequence and a gRNA targeting Rad52. The donor sequence containing the mMaple sequence was produced from plasmid pAT446 (pUM003, kindly provided by Jonas Ries laboratory, EMBL), using primers am1941 and am1942 (see *Supplementary file 2*). Cells were transformed simultaneously with the gRNA-RAD52 plasmid (pAT475) and the donor DNA on –LEU plates. For strains harboring Rad52-Halo, we amplified the HaloTag sequence from the plasmid pAT496 (pBS-SK-Halo-KanMX), kindly provided by the Carl Wu laboratory using primers am2182 and am2402. Cells were co-transformed with the gRNA-RAD52 plasmid (pAT475) and the donor DNA on –LEU plates. After clones validation by PCR (am384, am1944 and am1947), cells are re-streaked several times on YPD until the loss of the gRNA-RAD52 plasmid.

In Rad52-Halo strains, the pleiotropic drug resistance PDR5 gene was replaced by the N-acetyl-transferase (NAT) gene using deletion plasmid collection from the toolbox (*Janke et al., 2004*) and S1/S2 primers: pAT197, am2165 and am2166. The replacement of PDR5 gene with the 3-isopropyl-malate dehydrogenase (LEU2) gene was performed by PCR-based gene targeting using pAT297; primers am2609 and am2610. The PDR5 gene deletion was checked by PCR using following primers: am431, am2167, am2168 and am2421.

The functionality of Rad52 in these strains was verified using a dilution assay on plates containing MMS at different concentrations (*Figure 1—figure supplement 1*).

### Induction of a DSB

Before microscopy, strains harboring an I-*Sce*I cut site under galactose promoter were pre-grown in SC medium and diluted at $OD_{600nm}$ = 0.01 in SC +3% raffinose medium overnight. In the morning, cells were diluted at $OD_{600nm}$ = 0.2 in SC +3% raffinose medium until they reach $OD_{600nm}$ = 0.4. To induce the DSB, 2% galactose were then added directly into the tube for 2 hr, and fluorogenic JF646 (*Grimm et al., 2015*) were added directly into the tube during the last hour of galactose induction.

For the control experiments performed in the absence of DSB (referred as 'no DSB' in the figures), we used a strain without an I-*Sce*I cut site, grown in the same conditions as the strain used to induce a DSB to ensure a fair comparison between the two conditions.

### Sample preparation for the microscopy

A key aspect of fluorescent experiment is the choice of a suitable fluorophore. Several parameters have to be taken into account, depending on the experiments: specificity of the dyes, density of labeling, brightness and photo-stability. For SPT, in which we want to follow individual molecules as long as possible, we choose fluorogenic HaloTag JF646 dyes (*Grimm et al., 2015*) for their

brightness and specificity. Indeed, fluorogenic JF646 dyes have the ability to emit light only when they are bound to a Halo molecule and no unspecific signal was observed. Cells are prepared in dark tubes and incubated with 5 nM JF646 during 1 hr.

For live PALM, in which we want to observe each Rad52 molecules only once, we used a strain harboring endogenously fused Rad52-mMaple (*McEvoy et al., 2012*). Unlike the SPT experiments using external dyes, here, all the Rad52 molecules are fused to mMaple, allowing the observation of all Rad52 molecules for quantification of focus size, number of Rad52 molecules and density. Due to the photo-conversion properties of mMaple, it is estimated that 70% of mMaple can be photo-converted (*McEvoy et al., 2012*), which is taken into account in our analysis.

### *In vivo* tests of LLPS dissolution

To test the physical nature of Rad52 and Rfa1 foci, we used *in vivo* treatment known to disrupt LLPS but not solid aggregates. As described in *Kroschwald et al., 2017*, cells were treated with 1.6 hexanediol (Aldrich reference 240117) at 10% and digitonine (Merk Millipore reference 300410) at 0.001% for 30 min. Note that as reported in several studies (*Chang et al., 2018*; *Kroschwald et al., 2017*; *McSwiggen et al., 2019a*), extended exposure of cells to 1.6-hexanediol leads to loss of membrane integrity but a 30 min treatment does not alter membrane integrity (*Kroschwald et al., 2017*).

To compare the effect of fixation on Edc3, Rad52 and Rfa1 foci, fixation was performed during 10 min with 4% paraformaldehyde. We also tested several other fixation buffers (*Kaplan and Ewers, 2015*; *Schnell et al., 2012*, see fixation protocols below) but we found that Rad52 foci were not visible for these protocols.

### Detailed protocols fixation

> Protocol 1: 4% paraformaldehyde for 10 min, three rinsing steps in 2x SC media for 5 min each.
> Protocol 2: 4% paraformaldehyde for 10 min followed by 1% Glutaraldeide for 10 min, three rinsing steps in 2x SC media for 5 min each.
> Protocol 3: Methanol (à −20°C) for 10 min.
> Protocol 4: Ethanol (à −20°C) for 10 min.

All paraformaldehyde solutions were dissolved in the following fixation buffer: PFA 4%, 1X PBS, 2% sucrose.

### Single Particle Tracking

We acquired single molecule images on a custom setup based on a Nikon iSPT-PALM inverted microscope. For SPT experiments, we used fluorogenic JF646 dyes excited with a 647 nm laser with a power of 1.9 kW/cm$^2$ at the sample. The fluorescent signal is captured with an EM-CCD camera (Ixon Ultra 897 Andor) using a 100x/1.45NA (Nikon) objective. Using this objective, the image pixel size was 160 nm. We controlled the microscope with NIS software (Nikon). Experiments were performed at 30°C using a Tokai temperature control (STXG-TIZWX-SET) placed at the objective. We detected and connected the spots obtained in the movies with a custom algorithm derived from *Sergé et al., 2008* and we used home-made programs to visualize density and displacements detections. We kept all traces equal or longer than two points for further analysis.

### Estimation of diffusion coefficients

#### Displacement histogram

Probability Density Functions (PDF) are obtained after normalization of the displacement histograms and fitted with a 1-, 2-, or 3-population model (*Klein et al., 2019*; *Stracy and Kapanidis, 2017*):

1-population model: $PDF(r,t) = \frac{2r}{4Dt}\exp\left(\frac{-r^2}{4Dt}\right)$, where $D$ is the diffusion coefficient and $r$ is the position at time $t$.

2-population model: $PDF(r,t) = f\frac{2r}{4D_1t}\exp\left(\frac{-r^2}{4D_1t}\right) + (1-f)\frac{2r}{4D_2t}\exp\left(\frac{-r^2}{4D_2t}\right)$, where $D_1$ and $D_2$ are the diffusion coefficients of sub-population 1 and 2 respectively, and $f$ is the fraction of sub-population 1.

3-population model: $PDF(r,t) = f_1 \frac{2r}{4D_1 t} \exp\left(\frac{-r^2}{4D_1 t}\right) + f_2 \frac{2r}{4D_2 t} \exp\left(\frac{-r^2}{4D_2 t}\right) + (1 - f_1 - f_2) \frac{2r}{4D_3 t} \exp\left(\frac{-r^2}{4D_3 t}\right),$ where $D_1$, $D_2$ and $D_3$ are the diffusion coefficients of sub-population 1, 2, and, 3 respectively, and $f_1$ and $f_2$ are the fractions of sub-population 1 and 2.

In order to estimate the diffusion coefficients, we started out with the null hypothesis that traces were described by one diffusion coefficient. We simulated trajectories using the maximum likelihood estimate for the diffusion coefficient and compared the simulated and experimental CDF using a two-sided Kolmogorov-Smirnoff (KS) test to test the one diffusion coefficient hypothesis.

If one diffusion coefficient could not describe the data, we used maximum likelihood to find the two diffusion coefficients and the population fraction diffusing with each coefficient. We used a Metropolis Hastings algorithm to sample the two diffusion coefficients from the posterior distribution. Based on the distribution of the diffusion coefficients we estimated the uncertainty in the extracted parameters.

## Criteria to select traces inside repair foci

We use two methods to select traces inside foci:

i.  Since foci are much denser than the rest of the nucleus, we selected traces based on a density threshold. We first detect all the Rad52 molecules in our movies, and calculate a density map showing the number of Rad52 neighbours inside a sphere of 50 nm in radius (*Figures 1B* and *2B*, third panel). At this stage of the analysis, the tracking is not performed yet: we simply localize molecules and we have no information yet on Rad52 mobility. The focus is then determined by a density threshold as it corresponds to the region in the nucleus where repair proteins are highly concentrated. The program calculated the coordinates of a polygon, which defines the focus boundaries. Then, we perform the tracking step to connect Rad52 detections and draw a displacement map (*Figure 2B*, fourth panel): the traces inside a focus are obtained by overlying the polygon on the displacement map. For these traces, we tested a 2-population fit ($D_{inside-1}$ = 0.26 ± 0.02 µm²/s, and $D_{inside-2}$ = 0.066 ± 0.004 µm²/s, p=0.56, two-sided KS test), with slow molecules being the large majority (0.76 ± 0.04). However, the diffusion of fast molecules is similar to $D_2$ (0.24 ± 0.04), and by removing boundary points, we obtain a very good fit using a 1-population fit (p=0.99, two-sided KS test) inside the focus (data not shown). This analysis is referred as 'cropped data' in *Supplementary file 1*.

ii. Since no trace longer than 70 time-points were found in the absence of DSB, we selected traces longer than 70 time-points and assumed that they would be majorly populated by a slow diffusion coefficient. However, since molecules can travel back and forth between the focus and rest of the nucleus, we separated the traces belonging to the slow diffusion coefficient by assaying each displacement a probability based on the slow diffusion coefficient $D_3$ = 0.054 and removed points with probability less than 1/10 000. This analysis is refereed as 'Traces longer than 70 time-points' in *Supplementary file 1*.

After selecting traces inside foci with each method, we used MSD analysis to calculate the diffusion coefficient of molecules and decipher the nature of Rad52 motion inside foci (see below). Both methods lead to similar results (see *Supplementary file 1*).

## Mean square displacement

In the case of slow diffusion coefficients (D < 0.1 µm²/s), we needed to extract the diffusion coefficient from the experimental noise. For that, we calculated the MSD:

$$\mathrm{MSD}(n \cdot \Delta t) = \frac{1}{N-n} \sum_{i=1}^{N-n} \left[ (x_{i+n} - x_i)^2 + (y_{i+n} - y_i)^2 \right],$$

where $N$ is the number of points in the trajectory, (x, y) the coordinates of the locus in two-dimensions and $\Delta t$ the time interval used during the acquisition. To obtain a precise estimation of the confinement radius and the diffusion coefficient, we calculated time-ensemble-averaged MSD over several trajectories, which are simply referred to as 'MSD' in the Figures. To check possible artefacts due to the variability between molecules, we also computed individual MSD that is MSD calculated from a single trajectory (data not shown).

When the motion appeared confined, we fit the MSD with $\mathrm{MSD}(t) = R_\infty^2\left(1 - e^{-2dDt/R_\infty^2}\right) + 4\sigma^2$, to extract the experimental noise level $\sigma^2$, the radius of confinement $R_c = R_\infty\sqrt{\frac{2+d}{2}}$, where $d$ is the dimension of the motion, and the diffusion coefficient $D$ (*Klein et al., 2019*). The characteristic equilibration time after which the effect of boundaries appears is defined by the time at which the MSD curve starts bending and is given by $t_c = R_c^2/(6 \cdot D)$, where $R_c$ is the confinement radius and $D$ the diffusion coefficient inside foci (*Klein et al., 2019*).

When the motion is not a simple confinement but is modulated in time and space with scaling properties, it is called anomalous sub-diffusion. In this case, sub-diffusive loci are constrained, but, unlike confined loci, they can diffuse without boundaries and thus reach further targets if given enough time. For sub-diffusive motion, the MSD exhibits a power law and is fitted with:

$\mathrm{MSD}(t) = At^\alpha + \varepsilon$, where $\alpha$, the anomalous exponent, is smaller than 1, $A$ is the anomalous diffusion coefficient, and $\varepsilon$ is the noise due to the experimental measurements (*Miné-Hattab et al., 2017*, p.). For the wild type data, we fitted both functions to the data, and found a better agreement with the confinement fit.

## Estimation of the dynamic viscosity and the diffusion coefficient of Rad52 monomers *versus* multimers

To estimate the dynamic viscosity of the nucleus in the absence of DSB, we use Stokes-Einstein equation:

$$D = \frac{k_b T}{6\pi\eta r}$$

where $D$ is the diffusion coefficient, $r$ is the radius of a spherical molecule, $\eta$ is the dynamic viscosity, $k_B$ the Boltzmann constant and $T$ the absolute temperature.

We used 1.5 nm and 9 nm ±1 for the radius of a Rad52 monomer and the radius of a Rad52 multimer (*Saotome et al., 2018*; *Shinohara et al., 1998*).

Errors on $D$ and $\eta$ are calculated using error propagations: $\Delta\eta = \left|\frac{\partial D}{\partial \eta}\right|\Delta D$ and $\Delta D = \left|\frac{\partial \eta}{\partial D}\right|\Delta\eta$.

## Simulations

In order to extract as much information as possible from the traces, we performed simulations to correctly quantify the effects of: (1) the reduction in the number of observed fast molecules, due to their larger rate to exiting the z-frame undetected than slow molecules, and (2) the reduced diffusion coefficient, due to the collisions with the nucleus boundary.

To understand (1), we simulated 10000 particles for 10 s, inside a sphere of 1 μm radius, where traces were only measured if the particle within the window of |z| < 0.15 μm. We counted the number of detected data points, as a function of the size of the time window using time windows of ts = 5–100 ms. We found the number of detected data points decayed exponentially for all diffusion coefficients, but the magnitude of the decay rate grew as a function of the real diffusion coefficient. Therefore we took all of the computed decay rates, and fitted them to a power law, giving us an expression for the rate at which the number of observed molecules decays as a function of the time window and the true diffusion coefficient. We do not know the total number of particles in each population and we cannot determine this relation directly. We use this empirical scaling of the relation between the two populations.

In the same manner, we estimated (2) how the diffusion coefficients were affected. We measured the MSD and found that the diffusion coefficient obtained from this measure decayed as a function of the true diffusion coefficient. We directly calculated the diffusion coefficient from the maximum likelihood, and measured the gradient in the increase of the diffusion coefficient, assuming the considered time window is constant. By doing this we could fit the gradient and determine the expected diffusion coefficient.

## Estimation of radial motion

We used data cropped to movement inside the focus to estimate radial motion. We used all traces longer than 30 points and estimated the center of the focus by calculating their mean value for each

trace in the x-direction and y-direction. We calculated the distance to the center for all points, and defined the change in radius as Δr. We grouped data points from all traces and compared the statistics with a random walk with no attraction but with a similar diffusion coefficient (dashed lines *Figure 7D, E*).

## Live PALM

For live PALM experiments, we used strains harboring Rad52-mMaple as well as 1 or 2 I-*Sce*I cut-sites inducible under galactose. Samples were prepared with living cells and mMaple was excited with a 561 nm laser at power of 6 kW/cm$^2$ at the sample (max power) as well as a pulsed 405 nm laser to photo-activate mMaple. Movies were performed at 20 ms time interavals, during 2000 frames (no more spot were detectable at the end of the acquisition). Spots were detected using a custom algorithm derived from *Sergé et al., 2008* and home-made programs to visualize spots density and quantify foci.

# Acknowledgements

This work is dedicated to the memory of our friend and colleague Maxime Dahan. The authors thank Vu Nguyen and Carl Wu for fruitful exchanges on the use of JF dyes and for sharing plasmids. We thank Eric Coic for sharing with us a plasmid harboring Rad52-SUMO and to Siyu Liu for help in the revisions. We thank Antoine Coulon, Maxime Woringer, for their comments on the manuscript, as well as Bassam Hajj for his help during the construction of the PALM set-up. We also thank Mickael Garnier for his help on image analysis and Gaizka Le Goff for his help on preliminary experiments performed during his L3 internship. This work is at the origin of the Muse-IC project, a collaborative project between musicians and composers aiming to create musical pieces inspired by recent scientific discoveries (*Dolgin, 2019*).

AT team was financially supported by funding from the Labex DEEP (ANR-11-LABEX-0044 DEEP and ANR-10-IDEX-0001–02 PSL), and from the ANR DNA-Life (ANR-15-CE12-0007). AT and MD teams received support from the Fondation pour la Recherche Médicale (DEP20151234398); JMH from the ANR-12-PDOC- 0035–01. The AT team, TM and AMW were supported by Q-life (ANR-17-CONV-0005) and this project was also funded by the CNRS as part of its 80PRIME interdisciplinary programme. The authors greatly acknowledge the PICT-IBiSA@Pasteur Imaging Facility of the Institut Curie, member of the France Bioimaging National Infrastructure (ANR-10-INBS-04).

# Additional information

## Competing interests

Aleksandra M Walczak: Reviewing editor, *eLife*. The other authors declare that no competing interests exist.

## Funding

| Funder | Grant reference number | Author |
| --- | --- | --- |
| Agence Nationale de la Recherche | ANR-11-LABEX-0044 DEEP | Angela Taddei |
| Agence Nationale de la Recherche | ANR-10-IDEX-0001-02 PSL | Angela Taddei |
| Agence Nationale de la Recherche | ANR-15-CE12-0007 | Angela Taddei |
| Agence Nationale de la Recherche |  ANR-12-PDOC0035–01 | Judith Miné-Hattab |
| Fondation pour la Recherche Médicale | DEP20151234398 | Angela Taddei<br>Maxime Dahan |
| Agence Nationale de la Recherche | Q-life ANR-17-CONV-0005 | Judith Miné-Hattab<br>Thierry Mora<br>Aleksandra M Walczak |

| | | Angela Taddei |
| --- | --- | --- |
| Centre National de la Recherche Scientifique | 80—PRIME MITI project PhONeS | Judith Miné-Hattab Thierry Mora Aleksandra M Walczak Angela Taddei |

The funders had no role in study design, data collection and interpretation, or the decision to submit the work for publication.

### Author contributions

Judith Miné-Hattab, Conceptualization, Data curation, Software, Formal analysis, Supervision, Funding acquisition, Investigation, Methodology, Writing - original draft, Writing - review and editing; Mathias Heltberg, Software, Formal analysis, Validation, Investigation, Methodology, Writing - original draft; Marie Villemeur, Methodology, Design and construction of the strains; Chloé Guedj, Conceptualization, Methodology, Design and construction of the strains; Thierry Mora, Aleksandra M Walczak, Supervision, Funding acquisition, Validation, Investigation, Methodology, Writing - review and editing; Maxime Dahan, Conceptualization, Resources, Supervision, Funding acquisition, Validation, Investigation, Methodology; Angela Taddei, Conceptualization, Resources, Supervision, Funding acquisition, Validation, Investigation, Methodology, Project administration, Writing - review and editing

### Author ORCIDs

Judith Miné-Hattab (iD) https://orcid.org/0000-0001-9986-4092
Thierry Mora (iD) http://orcid.org/0000-0002-5456-9361
Aleksandra M Walczak (iD) http://orcid.org/0000-0002-2686-5702
Angela Taddei (iD) https://orcid.org/0000-0002-3217-0739

### Decision letter and Author response

Decision letter https://doi.org/10.7554/eLife.60577.sa1
Author response https://doi.org/10.7554/eLife.60577.sa2

# Additional files

### Supplementary files

• Supplementary file 1. Overview table of all the results This table provides the quantification of all the experiments performed, including statistical analysis of the results.

• Supplementary file 2. Yeast strains used in this study. This table provides the genotypes of the strains built and used for this study.

• Supplementary file 3. List of primers used in this study.

• Transparent reporting form

### Data availability

All data generated or analysed during this study are included in the manuscript and supporting files. Source data files are available on zenodo using the following link: https://zenodo.org/record/4495116.

The following dataset was generated:

| Author(s) | Year | Dataset title | Dataset URL | Database and Identifier |
| --- | --- | --- | --- | --- |
| Miné-Hattab J, Heltberg M, Villemeur M, Guedj C, Mora T, Walczak AM, Dahan M, Taddei A | 2021 | Single molecule microscopy reveals key physical features of repair foci in living cells | https://zenodo.org/record/4495116 | Zenodo, 10.5281/zenodo.4495116 |

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

## Appendix 1

### Consequences of increasing the concentration of Rad52

To evaluate the consequences of Rad52 concentration, we measured the concentration of Rad52 in the nucleoplasm for different levels of Rad52 over-expression. We built two haploid strains expressing Rad52 under the ADH or TEF1 promoters (5 and 12 times over-expression respectively) (*Hocher et al., 2018*). In addition, a single I-*Sce*I cut site allows the induction of a single DSB under the galactose promoter. Cells were grown at 25°C due to a slight growth defect of cells overexpressing Rad52. A single DSB was induced for 2 hr; during the last hour of induction, cells were incubated with JF646 dyes at 50 nM, a concentration 10 times higher than the one used for single molecule tracking. Such concentration allows the observation of the entire Rad52 focus as a single spot. Cells were observed on an inverted wide field microscope and Rad52 signal was quantified using a home-made software Q-foci (*Guidi et al., 2015*). Rad52 concentration in the nucleoplasm is calculated for each nucleus (nucleoplasm intensity divided by nucleoplasm volume).

We found that Rad52 intensity in the nucleoplasm is significantly different between wild type cells and cells expressing over-expressed level of Rad52 (see *Appendix 1—figure 1*, $p_{WT-ADH} = 6.5.10^{-8}$, $p_{ADH-TEF1} = 7.10^{-32}$ and $p_{WT-TEF1} = 3.5.10^{-8}$, Wilcoxon-Mann-Whitney test).

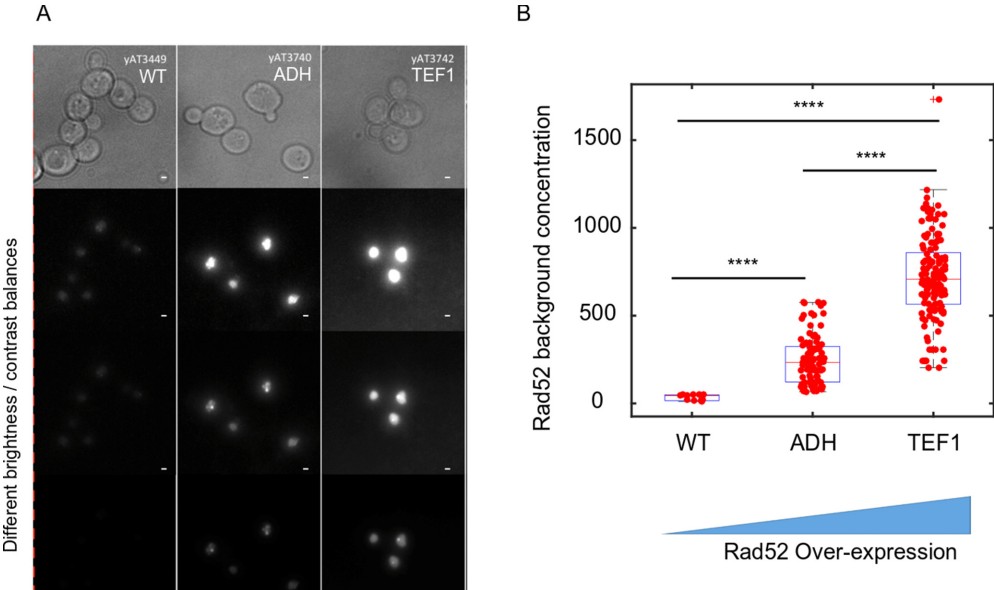

**Appendix 1—figure 1.** Consequences of increasing the concentration of Rad52 molecules in the cell. (**A**) Typical images of Rad52 after the induction a single DSB in wild type cells, and for two levels of Rad52 over-expression. The bar scale represents 1 μm. (**B**) Effect of Rad52 over-expression on the nucleoplasm concentration. 11, 94 and 141 cells were analyzed for wild type, ADH and TEF1 promoters respectively.

