## [Decision Letter]

**Acceptance summary:**

The revised manuscript shows elegant and compelling evidence that the dynamics of Rad52 molecules inside repair foci are consistent with the existence of a liquid droplet surrounding repair sites. Using single particle tracking (SPT) and Photo-activatable Localization Microscopy (PALM), the study also reveal that the ssDNA-binding protein RPA display a different behavior from Rad52, being mostly chromatin associated. Together, this analysis and the related conclusions significantly advance the field of nuclear dynamics in the context of DNA repair.

**Decision letter after peer review:**

Thank you for submitting your article "Single molecule microscopy reveals key physical features of repair foci in living cells" for consideration by *eLife*. Your article has been reviewed by three peer reviewers, and the evaluation has been overseen by a Reviewing Editor and Jessica Tyler as the Senior Editor. The reviewers have opted to remain anonymous.

The reviewers have discussed the reviews with one another and the Reviewing Editor has drafted this decision to help you prepare a revised submission.

As the editors have judged that your manuscript is of interest, but as described below that additional experiments are required before it is published, we would like to draw your attention to changes in our revision policy that we have made in response to COVID-19 (https://elifesciences.org/articles/57162). First, because many researchers have temporarily lost access to the labs, we will give authors as much time as they need to submit revised manuscripts. We are also offering, if you choose, to post the manuscript to bioRxiv (if it is not already there) along with this decision letter and a formal designation that the manuscript is "in revision at *eLife*". Please let us know if you would like to pursue this option.

Summary:

In this manuscript by Miné-Hattab and colleagues, the authors use single-molecule imaging approaches to investigate local dynamics of Rad52 foci at DSBs in budding yeast, which is an important area of investigation. They show that the dynamics of Rad52 molecules inside foci are consistent with protein movement within LLPS domains, while Rfa1 dynamics are not. Their data also provide supporting evidence to previous observations that repair sites cluster within the nuclei, and suggest that clustered foci behave as larger phase separated structures. While the idea that Rad52 and other repair proteins form phase separated domains is not novel, this study presents higher resolution data in support of this model. The reviewers generally agree that the study is interesting and well conducted, but the conceptual advancement is limited and a significant revision is needed for publication in *eLife*. Specifically, more convincing experiments demonstrating that the observed Rad52 dynamics reflect LLPS are required. Evidence that the dynamics are relevant for DNA repair and genome stability should also be provided. Additionally, the study should be better integrated with previous studies, statistical analyses need to be more rigorous/better presented, and a revisions of the text should include a clearer separation between observations and speculations.

The following is a summary of the main points brought up by the reviewers, including after consulting with each other.

Experimental revisions:

1) Additional data need to be provided to draw conclusions about whether or not the authors' observations are reflective of phase separation. Specifically, additional mobility studies in conditions that disrupt LLPS (chemical treatments and mutations) are needed, both for the individual protein and for the foci.

2) Regarding the possible categories of traces evaluated, one category is not included in the study. The surface tension that defines LLPS-dependent bodies is known to both help maintain focus integrity and partly counter LLPS body fusions. So if the foci represent true phase-separated bodies, have the authors then observed traces where Rad52 molecules interact with yet fail to enter the larger Rad52 foci?

3) How is it possible to distinguish a cluster of binding sites from liquid-liquid phase separation? In the absence of breaks, there are two Rad52 diffusion populations (D=1.2 and 0.3 um2/s), which the authors attribute to monomers and multimers. They don't verify these multimers by alternative approaches (say number and brightness analysis), but it seems like a reasonable possibility. After a break, a third component – slower than the previous two – becomes evident. This slow population coincides with the break. In the vicinity of the break, there is now only 1 component diffusion (D=0.03 um2/s). Also, the motion is now more confined, but not absolutely so. Also, Rad52 diffuses faster than Rfa1, which is bound to ssDNA. At this point, there is no data to distinguish between two possibilities: slow diffusion *or* diffusion + binding. Except, if it were diffusion + binding, one might perhaps expect to still see the free diffusion component.

The authors then turn to diffusion at the boundary (Figure 5), which I agree can be a more informative measure. Here, they see changes in the diffusion estimator for trajectories which cross the boundary, using displacement which they argue is more robust for slow diffusion. The problem is that the “boundary” is determined by the very thing they are trying to measure, not some independent marker of the compartment. In other words, Rad52 defines the compartment, unless I missed something fundamental in the experimental design. Ideally, the way such an experiment would be done to test the hypothesis that Rad52 is forming a LLPS compartment is to look at the diffusion of an inert tracer as it comes in and out of the compartment. As designed, I frankly do not see how the observation of different diffusivities in and out of the compartment distinguishes between a cluster of binding sites and an LLPS. If you accept that DNA-binding is in no way biasing the kinetics, then the authors' interpretation seems like the most sensible one. But the fact that Rad52 is involved in DNA repair makes that a hard assumption to swallow.

Furthermore, I'm not sure I entirely grasp the significance of Figure 6. Since Rad52 can easily escape one focus and enter another, regardless of whether it is a cluster of binding sites or a phase, I don't see how the radius of confinement measurement distinguishes between these two alternatives. The observation that the foci are 2x larger in diploids but at similar density is compelling, although recent data from the Brangwynne lab point out that conserved density need not be the case (PMID: 32405004).

4) In the syntax of this paper, Rad52 is a client in the LLPS, leaving the question of the scaffold unaddressed. After all, the Rad52 focus ultimately disappears, meaning that something caused this phase to be dispersed. So is RPA the scaffold? It might be possible to address both this point and point #3 knowing what is responsible for forming the LLPS in the first place.

Along the same lines, how do the authors reconcile previous findings indicating that recombinant DNA repair proteins phase separate in vitro with their claim that "Rad52 acts as a client of the LLPS but does not drive its formation"?

5) What is the evidence that the biophysical properties observed are of direct relevance to DNA repair? For example, is the mobility of Rad52 within the repair focus important for repair? Is the difference in diffusion kinetics within and outside of the repair focus important for genome stability? What could the authors do to alter that diffusion profile and what would be the consequence on repair? Also, addressing this point implies the need to use a more physiologically relevant system with repairable DSBs, and not the irreparable DSB system used here.

One should easily compare wild-type Rad52 to known Rad52 mutants (that partly or fully abrogate function) to see if there is any correlation between intra-focus mobility profiles and repair. In other words, experiments along these lines may indicate whether a particular intra-focus mobility profile consistently correlates with repair, while this profile is absent in non-functional mutants, or a particular type of mutant. These experiments should be very feasible.

6) Can the authors visualize the fusion of the Rad52 foci/DSBs in live cells within their experimental systems?

7) The statistical significance of most presented data is either lacking or unclear. This needs to be carefully addressed. Providing additional data files may also help the authors strengthen their findings.

Text revisions:

– Several statements made are not supported by the data and without clearly stating that the statements represent speculations. E.g. longer tail is due to Rad52 molecules diffusing slowly inside the focus; observing the 2 populations also in G1 does not necessarily mean that the 2 populations in S/G2 do not reflect replication forks at all. The authors need to carefully revise their claims/statements and consider alternative explanations. Also, the writing is often unclear or confusing and the authors should consider substantially revising it to clarify their claims, clearly indicate speculations that are not supported by the data, and make the text as accessible as possible to non-specialists.

– How was the cell cycle stage determined? This should be better explained.

– Figure 1—figure supplement 1 data appear to show the existence of a partial loss of Rad52 function in the Rad52-Halo cells. This should be clearly stated in the Results and consequent limitations/caveats discussed. Also, please clarify whether Figure 1—figure supplement 1 shows the viability of Rad52-Halo cells in the presence or absence of JF646.

– The authors present no direct evidence for an "attractive potential" that drives molecules towards the centre of the focus. For example, what if the “attractive potential” is simply the focus' boundary surface tension creating a barrier against which some of the molecules inside the focus bounce back towards the centre of the focus?

– The authors state that "Here, we found that upon different levels of Rad52 over-expression, the background concentration increases (Appendix 1—figure 1) suggesting that Rad52 might not be the driving molecule responsible for the LLPS formed at the damaged site." The logical transition here is unclear.

– It is difficult to judge the novelty of this work, as key papers that showed similar conclusions or datasets are not cited. Without direct comparison with other data sets, it is difficult to see exactly where this paper goes beyond published studies.

Here are a few key examples:

a) In the last year the Haber lab published a very similar study in Plos Genetics (Waterman et al., 2019). Although they tracked Ddc2 and Rad51, they also looked at the behavior of separate foci and this paper is not even cited. The data should be compared at the very least.

b) The characteristics of 53BP1 foci have been extensively studied by many labs including those of Altmeyer, Scherthan, DeLange and others, with very similar findings as Miné-Hattab reports for Rad52 (for example, Kilic et al., 2019; Sollazzo et al., 2018, as well as the single molecule work of the lab of Eric Greene). Moreover both rad52 and PCNA foci were studied by Essers et al. (Kanaar and Vermeulen) MCB 2005. 25(21): 9350-9359 and EMBO J. 2002 Apr 15. Comparisons with these studies needs to be made.

c) A number of earlier studies followed Rad52 foci in budding yeast on induced double strand breaks (even using the I-Sce1-cut system used here) that are not taken into consideration. The diffusion coefficients presented here have to be compared with these earlier studies and differences should be resolved by comparing techniques and conditions of imaging. For instance, Dion et al., 2012).

– It is unclear if the “absence of DNA damage” condition discussed in the first section of the Results is the non-induced version of the system described in the second section of the Results. Also regarding these sections, it seems that the “absence of DNA damage” control conditions were not conducted as part of the same experiments with the I-SceI DSB.

– Figure 2C is a little underwhelming. I would like to see one data panel in the main text with the cumulative distribution +/- DSB.

– "in the case of diffusion coefficients D<0.1 μm2/s, we use mean square displacement analysis allowing us to substrate the noise." Subtract the noise.

– One technical point. The fluorescence lifetime of a freely diffusing fluorophore (reported half-life of JF646 is 2.1 sec) is not the same as if it is bound to a protein; in the latter context the fluorophore would diffuse more slowly in the illumination path and be subjected to more rapid photobleaching, making 2.1 sec an overestimation of the half-life/underestimation of the decay rate. At a frame interval of 20ms, photobleaching may be a competitive rate for the disappearance of signal and I am not sure the lack of applying photobleaching correction or saying that the short tracks are not affected by that rate is properly justified.

---

## [Author Response]

Experimental revisions:1) Additional data need to be provided to draw conclusions about whether or not the authors' observations are reflective of phase separation. Specifically, additional mobility studies in conditions that disrupt LLPS (chemical treatments and mutations) are needed, both for the individual protein and for the foci.

We agree and thank the reviewer for this suggestion. In the new version of the manuscript, we now provide additional data in conditions that disrupt LLPS, for both Rad52 and Rfa1 foci. We used aliphatic alcohol hexanediol (1.6-hexanediol), a component proposed in the literature as a tool to differentiate between liquid-like and solid-like assemblies in living cells (Kroschwald et al., 2017, Strom et al., 2017, McSwiggen et al., 2019, Oshidari, 2020). In budding yeast, it has been shown that 1.6-hexanediol dissolves dynamic, liquid-like assemblies, such as P bodies, whereas solid-like assemblies, such as protein aggregates and cytoskeletal assemblies, are largely resistant to hexanediol (Kroschwald et al., 2017). We observed a clear difference between Rad52 and Rfa1 foci: following the induction of a single DSB, Rad52 foci were significantly less abundant in cells treated with hexanediol (8 % against 22 %) consistently with a recent report from the Meckhail laboratory (Oshidari et al., 2020), while Rfa1 foci remains stable. These results are presented and discussed in the revised manuscript (see Figure 4F and G and Figure 4—figure supplement 1).

In addition, we also tested the effect of 1.6-hexanediol at the single molecule level: we measured the mobility of individual Rad52 molecules (Rad52-Halo/JF646) following DSB induction and hexanediol treatment using the conditions in which the number of Rad52 foci is significantly decreased. However, comparing the histogram of Rad52 displacements in the nucleus with and without 30 minutes of hexanediol treatment, we did not observe any significant change in Rad52 mobility neither in the nucleoplasm nor in the few cells still harboring a Rad52 focus (2-sided Kolmogorov-Smirnoff test, p=0.995).

Finally, we also found that Rad52 foci are destabilized after fixation (10 minutes with 4% paraformaldehyde), as well as P bodies (Edc3 foci) described in the literature for their LLPS properties. However, Rfa1 foci resist to the same fixation treatment, suggesting that paraformaldehyde might not be able to fix LLPS (see Figure 4F and G and Figure 4—figure supplement 1).

2) Regarding the possible categories of traces evaluated, one category is not included in the study. The surface tension that defines LLPS-dependent bodies is known to both help maintain focus integrity and partly counter LLPS body fusions. So if the foci represent true phase-separated bodies, have the authors then observed traces where Rad52 molecules interact with yet fail to enter the larger Rad52 foci?

This is an excellent point and something we have investigated heavily, but this is something that is not possible for two reasons. First, the focus is diffusing as well, and therefore in order to find a specific trace at the boundary of the focus, one would need to know the exact position of the center of the focus, which due to movements can be shifted during the time steps considered. Secondly, and most importantly, the noise level in the experimental observations are of the order 25 nm, and since the estimated radius of the focus is around 100 nm, it is impossible to show if one particular trace has been repelled at the surface in the time between two successive observations. However statistically we have grouped observations based on their relative change in radial distance to the centre of the focus (this is shown in Figure 7). Here we have tried to measure exactly the presence of these events, but data-points of traces just outside the focus, are quite scarce and not something we have been able to draw any conclusions about.

3) How is it possible to distinguish a cluster of binding sites from liquid-liquid phase separation? In the absence of breaks, there are two Rad52 diffusion populations (D=1.2 and 0.3 um2/s), which the authors attribute to monomers and multimers. They don't verify these multimers by alternative approaches (say number and brightness analysis), but it seems like a reasonable possibility.

The principle of single molecule microscopy is to observe one molecule at a time, this molecule being bound to a single dye. In this study, we used flurogenic JF646 dyes (dye emitting light only once bound to Halo) that are not photo-activable. To reach the single molecule regime (as shown in Figure 1—figure supplement 2), we use an extremely low concentration of JF646 allowing us to obtain 0, 1 or maximum 2 detections per nucleus in the same frame. Thus, only a small fraction of Rad52-Halo molecules are bound to a JF646 dye and in case of multimer, the JF dye binds only 1 molecule of a multimer.

Thus, we cannot observe a difference in brightness: all detections have the same intensity regardless of whether they are monomeric or part of bigger multimeric complex.

After a break, a third component – slower than the previous two – becomes evident. This slow population coincides with the break. In the vicinity of the break, there is now only 1 component diffusion (D=0.03 um2/s). Also, the motion is now more confined, but not absolutely so. Also, Rad52 diffuses faster than Rfa1, which is bound to ssDNA. At this point, there is no data to distinguish between two possibilities: slow diffusion *or* diffusion + binding. Except, if it were diffusion + binding, one might perhaps expect to still see the free diffusion component.

We agree with the reviewer that in principle, our observations could be consistent with both hypotheses of “slow diffusion” or “diffusion + binding”. However, in the second case, we would expect to capture with SPT some intervals of time where Rad52 is bound. During these bound intervals, Rad52 should be immobile or diffuse slowly with the same coefficient as Rfa1. This would lead to a strongly multi-modal distribution of displacements inside foci, corresponding to 3 populations of freely-diffusing, bound, and mixed states (intervals in which the particle may bind or unbind). We do not observe such multi-modal behavior at our resolution of 20 ms intervals. This does not rule out “diffusion + binding”, but it puts hard constraints on the rates of binding and unbinding, which must be much faster than 20ms to explain the smoothness of the displacement histogram.

We can try to quantify this by making the conservative assumption that binding is diffusion limited, with diffusivity D = 1 μm^2^/s, and a linear binding size a = 10 nm. In that case the binding rate of individual molecules to a given target is k_on_ = 4Da (Berg and Purcell, 1977). Assuming a realistic range of binding dissociation constants K_d_ ~ 1nM to 1μM and using the relation K_d_ = k_off_ / k_on_, this implies a mean bound time of 1/k_off_ = 1/K_d_/k_on_ ~ 40 ms to 40 s, values which are all larger than our resolution. In fact, to explain the smoothness of our displacement histogram, the binding specificity would have to be very weak, Kd >> 2 μM, which is far above values observed in vivo.

In the revised version of the manuscript, we now discuss this point.

The authors then turn to diffusion at the boundary (Figure 5), which I agree can be a more informative measure. Here, they see changes in the diffusion estimator for trajectories which cross the boundary, using displacement which they argue is more robust for slow diffusion. The problem is that the “boundary” is determined by the very thing they are trying to measure, not some independent marker of the compartment. In other words, Rad52 defines the compartment, unless I missed something fundamental in the experimental design. Ideally, the way such an experiment would be done to test the hypothesis that Rad52 is forming a LLPS compartment is to look at the diffusion of an inert tracer as it comes in and out of the compartment. As designed, I frankly do not see how the observation of different diffusivities in and out of the compartment distinguishes between a cluster of binding sites and an LLPS. If you accept that DNA-binding is in no way biasing the kinetics, then the authors' interpretation seems like the most sensible one. But the fact that Rad52 is involved in DNA repair makes that a hard assumption to swallow.

We thank the reviewer for this remark because it is an important point to clarify. A repair focus is defined by a high concentration of repair proteins at a site of DNA damage. In our experiments, we first detect all the Rad52 molecules in our videos, and calculate a density map showing the number of Rad52 neighbours inside a sphere of 50 nm in radius (Figure 1B and 2B, third panels). At this stage of the analysis, the tracking is not performed yet: we simply localize molecules and we have no information yet on Rad52 mobility. The focus is then determined by a density threshold as it corresponds to the region in the nucleus where repair proteins are highly concentrated. The program calculated the coordinates of a polygon, which defines the focus boundaries. Then, we realized the tracking step to connect Rad52 detections and draw a displacement map (Figure 2B, forth panel): the traces inside a focus are obtained by overlying the polygon on the displacement map. Thus, in this method, the determination of the focus is independent of Rad52 mobility. We then found that inside foci, Rad52 mobility is slower, and that traces crossing the focus boundary sharply change mobility as they come in and out the sub-compartment. We now clarify the method (Figure 2—figure supplement 2).

To estimate the mobility of Rad52 molecules within foci, we also used a second method to identify the traces inside foci: since no trace longer than 70 time-points were found in the absence of DSB, we selected traces longer than 70 time-points and define these ones as the ones belonging to the focus. Both methods give similar distribution of step sizes and fitting the probability density function leads to the similar diffusion coefficient, as shown in Figure 2—figure supplement 2.

We agree that looking at the diffusion of an inert tracer as it comes in and out of the compartment would be interesting. However, we have not found yet an inert tracer that penetrates into repair foci and that we can follow at the single molecule level. Further assays would need to be developed to go in this direction.

Furthermore, I'm not sure I entirely grasp the significance of Figure 6. Since Rad52 can easily escape one focus and enter another, regardless of whether it is a cluster of binding sites or a phase, I don't see how the radius of confinement measurement distinguishes between these two alternatives.

Thank you for this comment. We agree that this point needs clarifying. In a simplified view where multiple DSBs cluster and stay close without merging, we expect to observe the same confinement radius than following a single DSB. In another extreme view, multiple DSBs fuse, giving rise to a large single droplet, and Rad52 confinement radius increases.

Since exchange is important, we now discuss a third intermediate view in which DSBs cluster without merging, but we include a high rate of exchange making possible for a Rad52 molecule to “jump” between a focus to another one. In this view, we should observe bimodal trajectories, exhibiting confined diffusion in the first focus, followed by a confined diffusion in the second focus. Such traces should be observable since the mean residence inside a focus is 240 ms (12 frames) and traces are often longer than this (we added the histogram of Rad52 traces length inside foci, see new Figure 5—figure supplement 1).

After examining carefully individual trajectories inside a focus induced by 2 DSBs, we never observe bimodal trajectories. Instead, in typical trajectories (Figure 6F, right panel), Rad52 is homogenously distributed in time within a larger focus. In the revised version of the manuscript, we now comment on this point. In the limit case where the two foci are so close that we can’t visualize jumps the second and third views are effectively the same.

The observation that the foci are 2x larger in diploids but at similar density is compelling, although recent data from the Brangwynne lab point out that conserved density need not be the case (PMID: 32405004).

First, we would like to clarify this point: in Figure 6, we observed that foci induced by 2 DSBs are 2x larger than those induced by a single DSB. We have not performed experiments in diploid cells in the first version of the manuscript.

Then, we explain that 2 droplets of radius R would lead to a bigger droplet of doubled volume when they fuse, with no significant difference in density (see p 13). We thank the reviewer for bringing our attention to the recent study by the Brangwynne lab. In the experiment following 2 DSBs, there is no change of Rad52 concentration in the nucleus, so we believe the recent study by the Brangwynne lab is of no direct relevance for these results. However, this study led us to revisit our results following Rad52 over-expression. We now comment this point.

4) In the syntax of this paper, Rad52 is a client in the LLPS, leaving the question of the scaffold unaddressed. After all, the Rad52 focus ultimately disappears, meaning that something caused this phase to be dispersed. So is RPA the scaffold? It might be possible to address both this point and point #3 knowing what is responsible for forming the LLPS in the first place.

We do not know what is the scaffold of repair foci but it could not be Rfa1 because one expects the scaffold to form a LLPS, unlike Rfa1.

Along the same lines, how do the authors reconcile previous findings indicating that recombinant DNA repair proteins phase separate in vitro with their claim that "Rad52 acts as a client of the LLPS but does not drive its formation"?

We thank the reviewer for this interesting remark. Recent studies have shown that the physical nature of proteins complexes can differ from in vitro to in vivo assays. In a recent publication on the nature of heterochromatin (Erdel *et al.,* 2020), the authors confirm that the HP1α can form droplets in vitro as also observed by Larson *et al.*, Nature 2018. However, HP1α droplets formation in vitro requires a high protein concentration (> 40 μM): it is not certain that the in vitro behavior of highly concentrated and phosphorylated HP1 accurately recapitulates the in vivo biology with physiological HP1 level (~ 1 μM). Indeed, the results in vivo presented by Erdel et al., 2020, clash with a simple LLPS mechanism for heterochromatin subcompartments; instead, they are more consistent with a collapsed “chromatin-globule model” (or polymer-polymer phase separation model, PPPS).

As underlined in the recent literature (Gitler *et al.,* Mol Cell 2020 Previews, McSwiggen *et al.,* 2019), with the advances in super resolution microscopy and biophysical approaches, what appear to be droplets at first glance might in fact follow more complex models.

Taking into consideration the work from the Brangwynne lab showing that LLPS composed by multiple proteins with heterotypic multicomponent interactions do not exhibit a fixed saturation concentration, we propose that the formation of repair foci could be driven by a combination of several interacting proteins including Rad52. We now comment on this point.

5) What is the evidence that the biophysical properties observed are of direct relevance to DNA repair? For example, is the mobility of Rad52 within the repair focus important for repair? Is the difference in diffusion kinetics within and outside of the repair focus important for genome stability? What could the authors do to alter that diffusion profile and what would be the consequence on repair? Also, addressing this point implies the need to use a more physiologically relevant system with repairable DSBs, and not the irreparable DSB system used here.

Thank you for the remark and we agree with the reviewer’s comment. To verify if our observations are conserved in a system with repairable DSBs, we measured Rad52 mobility in diploids cells, after the induction of a single DSB at the same locus as in haploids (*LYS2*). We obtained results very similar to the one obtained in haploid cells: with Rad52 molecules exhibiting: (i) 3 populations, (ii) confined motion inside foci, (iii) a difference in diffusion coefficient of 8 between internal Rad52 mobility inside foci and the whole focus mobility. It is important to note that all our experiments in haploid cells where performed 2 hours after induction which correspond to the step of search for an ectopic homologous sequence (Piazza et al., 2019). This could explain why Rad52 has the same behavior in the presence and in the absence of a donor sequence.

We now present the results in Figure 2—figure supplement 5, and we discuss this point in the text.

One should easily compare wild-type Rad52 to known Rad52 mutants (that partly or fully abrogate function) to see if there is any correlation between intra-focus mobility profiles and repair. In other words, experiments along these lines may indicate whether a particular intra-focus mobility profile consistently correlates with repair, while this profile is absent in non-functional mutants, or a particular type of mutant. These experiments should be very feasible.

It will of course be interesting to combine SPT and genetics to test to which extent repair proteins mobility, residence time inside foci internal dynamics and type of motion are modified when DNA repair is deficient. In the revised manuscript, we show that Rad52-SOMUylation affect some of the observables accessible by SPT (Figure 2, Figure 2—figure supplement 3,). We agree that testing different Rad52 mutant will be very interesting but this will be the subject of another manuscript.

6) Can the authors visualize the fusion of the Rad52 foci/DSBs in live cells within their experimental systems?

The fusion of repair foci has been already observed both in yeast (Oshidari et al., 2020 and Waterman et al., 2019) and in human cells (Altmeyer et al., 2015) by tracking repair foci at larger time scales (several minutes). We now comment this point in the Discussion. Here we focused on foci size and internal dynamics inside foci in response to 1 *versus* 2 DSBs, which brings new information that are only accessible by single molecule microscopy.

7) The statistical significance of most presented data is either lacking or unclear. This needs to be carefully addressed. Providing additional data files may also help the authors strengthen their findings.

Thank you for this remark. We have added statistical tests throughout the revised manuscript. To be more specific, we added:

– goodness of fit for all the MSD fit, indicating the Pearson's chi-squared (see revised Table 1)

– the p-values of the Wilcoxon-Mann-Whitney test for the data with over-expressed levels of Rad52 (Figure 6—figure supplement 1)

Text revisions:– Several statements made are not supported by the data and without clearly stating that the statements represent speculations. E.g. longer tail is due to Rad52 molecules diffusing slowly inside the focus; observing the 2 populations also in G1 does not necessarily mean that the 2 populations in S/G2 do not reflect replication forks at all.

We now modulate our interpretation of the 2 populations in S/G2.

The authors need to carefully revise their claims/statements and consider alternative explanations. Also, the writing is often unclear or confusing and the authors should consider substantially revising it to clarify their claims, clearly indicate speculations that are not supported by the data, and make the text as accessible as possible to non-specialists.

In the revised version, we made the text more accessible, for example,

– we moved in the method or figure legend some technical details explaining how the diffusion coefficient is estimated inside repair foci (see Results and legend of Figure 2—figure supplement 2).

– We explained the distribution of traces length in a more accessible manner.

– We removed some redundancies in the Discussion and discuss the models of sub-compartments in a more accessible manner.

– How was the cell cycle stage determined? This should be better explained.

Cell cycle stage was assessed by the presence and the size of a bud on the transmission image. We now clarified this point in the text (figure legend of Figure 1).

– Figure 1—figure supplement 1 data appear to show the existence of a partial loss of Rad52 function in the Rad52-Halo cells. This should be clearly stated in the Results and consequent limitations/caveats discussed. Also, please clarify whether Figure 1—figure supplement 1 shows the viability of Rad52-Halo cells in the presence or absence of JF646.

We have commented on this point in the legend of Figure 1—figure supplement 1.

– The authors present no direct evidence for an "attractive potential" that drives molecules towards the centre of the focus. For example, what if the “attractive potential” is simply the focus' boundary surface tension creating a barrier against which some of the molecules inside the focus bounce back towards the centre of the focus?

We thank the reviewer for this comment that definitely needs clarification. When we refer to an attractive potential, we do not mean a potential that attracts the molecules towards the center of the focus, but a potential that, exactly like it would be the case for surface tension, hinders the passage out of the focus for the molecules. To clarify this important passage, we have updated the text, in the last paragraph of the Results part, following text for Figure 7.

– The authors state that "Here, we found that upon different levels of Rad52 over-expression, the background concentration increases (Appendix 1—figure 1) suggesting that Rad52 might not be the driving molecule responsible for the LLPS formed at the damaged site." The logical transition here is unclear.

We have clarified this point and added a reference.

– It is difficult to judge the novelty of this work, as key papers that showed similar conclusions or datasets are not cited. Without direct comparison with other data sets, it is difficult to see exactly where this paper goes beyond published studies.Here are a few key examples:a) In the last year the Haber lab published a very similar study in Plos Genetics (Waterman et al., 2019). Although they tracked Ddc2 and Rad51, they also looked at the behavior of separate foci and this paper is not even cited. The data should be compared at the very least.

The work by Waterman et al., 2019 is actually different from our present work although complementary. Waterman et al. did not monitor the dynamics of single molecules but the dynamics of repair foci decorated either by Rad51 or Ddc2 in a strain experiencing 3 DSB. They monitored the aggregation and mobility of DSBs into a single focus and show that these events were independent of Rad52.

This is paper is now cited in the revised version.

b) The characteristics of 53BP1 foci have been extensively studied by many labs including those of Altmeyer, Scherthan, DeLange and others, with very similar findings as Miné-Hattab reports for Rad52 (for example, Kilic et al., 2019; Sollazzo et al., 2018, as well as the single molecule work of the lab of Eric Greene). Moreover both rad52 and PCNA foci were studied by Essers et al. (Kanaar and Vermeulen) MCB 2005. 25(21): 9350-9359 and EMBO J. 2002 Apr 15. Comparisons with these studies needs to be made.

Thank you, we added these references and discuss them in the revised manuscript.

c) A number of earlier studies followed Rad52 foci in budding yeast on induced double strand breaks (even using the I-Sce1-cut system used here) that are not taken into consideration. The diffusion coefficients presented here have to be compared with these earlier studies and differences should be resolved by comparing techniques and conditions of imaging. For instance, Dion et al., 2012).

We now compare the diffusion coefficient of the whole focus measured here with the one obtained in Dion et al., 2012.

– It is unclear if the “absence of DNA damage” condition discussed in the first section of the Results is the non-induced version of the system described in the second section of the Results.

The “absence of DNA damage” condition is indeed the same experiment described in the second section of the Results and referred as “no DSB” in the figures. For this condition, we used a strain without a I-*Sce*I cut site and grown in the same conditions as the strain used to induce a DSB to ensure a fair comparison between the two conditions (with or without DSB). We now clarify this point in the Materials and methods section.

Also regarding these sections, it seems that the “absence of DNA damage” control conditions were not conducted as part of the same experiments with the I-SceI DSB.

As mentioned above, experiments performed in the absence or in the presence of DNA damage uses a different strain and were done separately. We have repeated these experiments 3 to 6 times, and one set of data includes the experiments with and without DSB done the same day. We also use the “absence of DNA damage” condition as a control experiment for the stability of our set-up: this experiment has been systematically performed and analysed once a year and similar diffusion coefficients were obtained.

– Figure 2C is a little underwhelming. I would like to see one data panel in the main text with the cumulative distribution +/- DSB.

We now present the cumulative distribution of traces length for Rad52-Halo/JF646, in the absence and in the presence of a DSB (see new Figure 2—figure supplement 1).

– "in the case of diffusion coefficients D<0.1 μm2/s, we use mean square displacement analysis allowing us to substrate the noise." Subtract the noise.

We made the correction.

– One technical point. The fluorescence lifetime of a freely diffusing fluorophore (reported half-life of JF646 is 2.1 sec) is not the same as if it is bound to a protein; in the latter context the fluorophore would diffuse more slowly in the illumination path and be subjected to more rapid photobleaching, making 2.1 sec an overestimation of the half-life/underestimation of the decay rate. At a frame interval of 20ms, photobleaching may be a competitive rate for the disappearance of signal and I am not sure the lack of applying photobleaching correction or saying that the short tracks are not affected by that rate is properly justified.

We thank the reviewer for this remark. The JF646 used in this study are flurogenic, meaning that they emit light only when they are bound to a Halo molecule. Thus, there is no signal when free JF646 dyes diffuse inside a cell that does not express Halo. To measure the half life time of JF646, we used the strain harbouring Rad52-Halo, in the absence of DSB and we incubated the JF646 with the cells for 1h for 50 nM (a concentration 10 times higher than the one used to be in the single molecule regime). This point is explained in Figure 1—figure supplement 3.